



# Accuracy and precision of lower stratospheric polar reanalysis temperatures evaluated from A-train CALIOP and MLS, COSMIC GPS RO, and the equilibrium thermodynamics of supercooled ternary solutions and ice clouds

Alyn Lambert[1] and Michelle L. Santee[1]

[1]Jet Propulsion Laboratory, California Institute of Technology, Pasadena, California, USA

*Correspondence to:* A. Lambert, (Alyn.Lambert@jpl.nasa.gov)

**Abstract.**

We investigate the accuracy and precision of polar lower stratospheric temperatures (100–10 hPa during 2008–2013) reported in several contemporary reanalysis data sets comprising two versions of the Modern-Era Retrospective analysis for Research and Applications (MERRA and MERRA-2), the Japanese 55-year Reanalysis (JRA-55), the European Centre for
Medium-range Weather Forecasts (ECMWF) interim reanalysis (ERA-Interim), and the National Oceanic and Atmospheric Administration (NOAA) National Centers for Environmental Prediction (NCEP) Climate Forecast System Reanalysis (NCEP-CFSR). We also include the Goddard Earth Observing System Model version 5.9.1 near real-time analysis (GEOS-5.9.1). Comparisons of these datasets are made with respect to retrieved temperatures from the Aura Microwave Limb Sounder (MLS), Constellation Observing System for Meteorology, Ionosphere and Climate (COSMIC) Global Positioning System (GPS) Radio
Occultation (RO) temperatures, and independent absolute temperature references defined by the equilibrium thermodynamics of supercooled ternary solutions (STS) and ice clouds. Cloud-Aerosol Lidar with Orthogonal Polarization (CALIOP) observations of polar stratospheric clouds are used to determine the cloud particle types within the Aura MLS geometric field of view. The thermodynamic calculations for STS and the ice frost point use the colocated MLS gas-phase measurements of $HNO_3$ and $H_2O$. The estimated accuracy and precision for the STS temperature reference, over the 68 to 21 hPa pressure range, is
0.6–1.5 K and 0.3–0.6 K, respectively; for the ice temperature reference they are 0.4 K and 0.3 K, respectively. These uncertainties are smaller than those estimated for the retrieved MLS temperatures and also comparable to GPS RO uncertainties (accuracy<0.2 K, precision >0.7 K) in the same pressure range.

We examine a case study of the time-varying temperature structure associated with layered ice clouds formed by orographic gravity waves forced by flow over the Palmer peninsula, and compare how the wave amplitudes are reproduced by each reanal-
ysis data set. We find that the spatial and temporal distribution of temperatures below the ice frost point, and hence the potential to form ice PSCs in model studies driven by the reanalyses, varies significantly because of the underlying differences in the representation of mountain wave activity. We have therefore used temperature variances, calculated from the COSMIC GPS RO temperature data (80–20 hPa), and imposed a variance threshold to selectively remove profiles with suspected enhanced wave activity. We examine the resulting improvement to the fidelity of the reanalysis temperatures.





High accuracy COSMIC temperatures are used as a common reference to intercompare the reanalysis temperatures and, although the COSMIC data are routinely assimilated by each reanalysis scheme except for MERRA, we find that there are significant departures from uniformity in the structure of the meridional and altitude temperature differences. Over the 68–21 hPa pressure range, the biases of the reanalyses with respect to COSMIC temperatures for both polar regions fall within the narrow range of −0.6 K to +0.5 K. The corresponding standard deviations of the differences are ∼0.8 K at 100 hPa and increase exponentially with altitude, as expected because of the gradually worsening GPS RO measurement precision. GEOS-5.9.1, MERRA, MERRA-2 and JRA-55 have predominantly cold biases, whereas ERA-I has a predominantly warm bias. NCEP-CFSR has a warm bias in the Arctic, but becomes substantially colder in the Antarctic.

For the comparisons of the reanalysis temperatures with the thermodynamically calculated temperature references, we use the concept of an instrument field of view (FOV) fill-fraction. The distribution of CALIOP PSC types within the MLS geometric FOV are used to mitigate the effects of disparate types of PSCs occurring in the much larger MLS sample volume. This removes the speckle effect that arises either because of uncertainty in the detection classification from random noise in the CALIOP signals or from the microphysical effects of small-scale temperature fluctuations on the formation of PSC particles. We select viewing scenes with the requirement that 75% or more of the MLS geometric FOV is filled with CALIOP PSC detections of the same PSC type classification. Scenes satisfying this requirement for CALIOP STS detections we denote as LIQ, and for CALIOP ice detections we denote as ICE. Over the 68–21 hPa pressure range, the reanalysis temperature biases are in the range −1.6 K to −0.3 K with standard deviations ∼0.6 K for the LIQ reference, and in the range −0.9 K to +0.1 K with standard deviations ∼0.7 K for the ICE reference. Comparisons of MLS temperatures with the LIQ and ICE reference temperatures reveal vertical oscillations in the MLS temperatures, and a significant low bias in MLS temperatures of up to 3 K.

# 1 Introduction

Over the last couple of decades, global reanalysis datasets have become one of the workhorse tools of the climate research community for understanding atmospheric processes and variability (Fujiwara et al., 2017) and more recently for potentially investigating climate changes (Thorne and Vose, 2010; Dee et al., 2014; Simmons et al., 2014). A reanalysis system combines observations with predictions from a global forecast model that propagates information forward in time and space using an assimilation scheme to produce a control-weighted blend of the observations and the new forecast. Unlike operational analysis schemes, which are updated as needed to improve numerical weather prediction (NWP) capabilities, reanalysis systems are designed to be conservative and retain the same code-base, with the aim of producing consistent and low-artifact output content over their entire multi-decade timeseries for a given product version generation.



The state-of-the-art of NWP and reanalysis data has improved greatly over the years as computational technology has evolved and new observation systems, such as global navigation satellite system (GNSS) radio occultation (RO) and data from other advanced satellites (including one-off research satellites), have also been brought within the realm of data assimilation. Currently, there are numerous NWP centers that produce reanalysis products. Understandably, there are differences in the tech-

nical implementation of the highly complex reanalysis schemes between the NWP centers, and these can lead to differences in the reanalysis products. Accordingly, there is a need for the research community to know how the deficiencies of a particular reanalysis may impact their investigations. The SPARC (Stratosphere-troposphere Processes And their Role in Climate) Reanalysis Intercomparison Project (S-RIP) (Fujiwara et al., 2017) is a coordinated activity with the aim of understanding the underlying causes of differences among global reanalysis data products to help support the needs of the research commu-

nity. Comparisons of reanalysis data with independent observations form an indispensable part of this assessment, and a core component of the evaluation process is to use satellite observations as a reference.

The focus of this paper is on the "Polar Processes" theme of the upcoming associated S-RIP report outlined by Fujiwara et al. (2017), in particular the intercomparison of reanalysis temperatures, and we seek to answer the question *"Can the thermodynamics of supercooled ternary solutions and the ice frost point be used to provide an absolute temperature reference*

*in the polar regions?"*. Polar processes, such as the potential for ozone loss via heterogeneous reactions (Solomon, 1999; Solomon et al., 2015), depend critically on temperature. The potential ozone loss is related to the volume of stratospheric air that is below certain critical temperature thresholds associated with formation of PSCs (Rex et al., 2004; Orsolini et al., 2009; Harris et al., 2010). Mean winter temperatures in the Arctic stratosphere are significantly higher than those in the Antarctic (e.g. Waugh and Polvani, 2010), such that the propensity for the formation of persistent Arctic PSCs is much lower, and the

development of widespread synoptic ice PSCs is quite rare (Engel et al., 2013). Since the results of polar modeling studies are especially susceptible to errors in the underlying temperature fields, Chemistry-Climate Models (CCMs) (Butchart et al., 2011) are preferably used in a specified-dynamics (SD) mode for studying polar processes. In the SD-mode, CCMs are constrained by using Newtonian relaxation to nudge the model output at each time step towards the reanalysis temperatures and wind fields (e.g. Whole Atmosphere Community Climate Model (WACCM), Wegner et al., 2013)), and are therefore adapted better

to represent polar winters with unusual characteritics such as split vortices, exceptionally cold conditions, sudden warmings, etc. Accurate temperatures are critical, not only to process studies, but also to assessment of trends in stratospheric winter conditions, especially considering the prospects of an increased frequency of future episodes of severe Arctic ozone depletion (Rex et al., 2006; Sinnhuber et al., 2011; Langematz et al., 2014), following the record low seen in 2011 (Manney et al., 2011). A temperature difference of less than 1 K can have a significant effect on the outcome of model runs that are driven

by reanalysis data. Errors in the extent of heterogeneous processing, arising from temperature biases, result in under/over estimate of activated chlorine, thus leading to too little/much chemical ozone loss. Temperature adjustments have often been applied in order to better match model predictions with observations (e.g. Danilin et al., 2000). Wohltmann et al. (2013) found a discrepancy in modeled and observed HCl and ClO which could be improved by adjusting the reanalysis temperatures by $-1.0$ K for ERA-I in the Arctic 2009/2010 winter. Brakebusch et al. (2013) found an improvement in modeled ozone loss

with a $-1.5$ K adjustment for an unspecified version of GEOS5 in the Arctic 2004/2005 winter. Wegner et al. (2012) showed a



comparison of ERA-I Arctic temperatures in March 2005 that indicates a 1.5 K warm bias in ERA-I below 205 K compared to the temperatures measured with the Geophysica. Solomon et al. (2015) demonstrated that a $-2.0$ K adjustment to the MERRA temperatures in the 2011 Arctic winter provides a better fit to the zonally averaged total column ozone. Improved knowledge of the temperature biases in reanalysis data would enhance confidence in the attribution of model errors to the underlying

physico-chemical properties of PSCs and heterogeneous reactions.

Several previous intercomparisons of analyses and reanalyses generated by various national centers have been carried out (e.g. Manney et al., 1996; Pawson et al., 1999; Manney et al., 2003, 2005) to assess their accuracy and ultimate suitability for polar studies. Independent datasets such as radiosondes, satellite observations, Global Positioning System (GPS) Radio Occultation (RO) (Nedoluha et al., 2007) and temperature sensors on long-duration balloons (Hertzog et al., 2004; McDonald

and Hertzog, 2008) have also been used to assess reanalysis temperatures.

Historically, the uncertainties in polar reanalysis temperatures, especially for the southern hemisphere, have been higher than those in other regions of the globe, because of sparse coverage from conventional temperature measurements such as radiosondes (e.g. Gobiet et al., 2005; de la Torre Juárez et al., 2009). In recent decades, the augmentation of the coverage of polar regions by an increase in satellite missions, including assimilation of research satellite data, and notably the advent of

GNSS RO, has dramatically improved the situation (Wang and Lin, 2007). Lawrence et al. (2015) examined a 34-year record of polar processing diagnostics for MERRA and ERA-Interim reanalyses. They documented the introduction of new data streams and noted better agreement in the post-2001 timeframe following the assimilation of Aqua Atmospheric Infrared Sounder (AIRS) and Geostationary Operational Environmental Satellite (GOES) radiances into both schemes.

We confine our investigations to the temperature data over the past decade from six contemporary reanalysis datasets and

introduce a novel analysis based on the thermodynamics of supercooled ternary solutions and the ice frost point to provide an absolute temperature reference. Near-simultaneous and colocated measurements of nitric acid, water vapor and cloud phases are currently only afforded by the precise formation flying of two satellite instruments in the A-train. These are the Aura Microwave Limb Sounder (MLS) instrument, which measures the gas-phase species, and the Cloud-Aerosol Lidar with Orthogonal Polarization (CALIOP) lidar, which is used to classify PSC types. The analysis presented refines and extends the

methodology used originally by Lambert et al. (2012) to investigate the temperature existence regimes of different types of PSCs. Supercooled Ternary Solution (STS) and ice PSCs are identified by the CALIOP lidar PSC classification and combined with spatially and temporally colocated MLS $HNO_3$ and $H_2O$ gas-phase abundances. We accumulate statistics on the existence regimes for STS and ice PSCs by using CALIOP to identify the presence of PSCs in the near-simultaneous and colocated MLS geometric field of view at the along track resolution (165 km by 2.16 km). MLS is used to obtain the spatially and temporally

colocated ambient gas-phase $H_2O$ and $HNO_3$ volume mixing ratios. These are required to calculate the theoretical equilibrium temperature dependence of the STS ($T_{eq}$) and ice ($T_{ice}$) PSCs. We compare the observed and calculated temperature distributions of (a) the uptake of $HNO_3$ in STS, and (b) the ice frost point, for each reanalysis data set and for MLS temperature. However, lack of sufficient statistics in the Arctic precludes a robust conclusion for ICE PSCs. We also compare the Antarctic and Arctic reanalysis temperatures poleward of 60° with the COSMIC temperatures used as a common reference (Section 2.4).

We investigate six Antarctic PSC seasons from 20 May (d140) to 18 August (d230) from 2008 to 2013 in the lower stratosphere





(100 hPa–10 hPa) for latitudes poleward of 60° S. Similarly, in the Arctic we investigate from 2 December (d336) to 31 March (d090) from 2008/2009 to 2012/2013 for latitudes poleward of 60° N.

In section 2 we review the satellite instruments, reanalysis datasets, and methodology. In section 3 we review the equilibrium thermodynamics associated with the formation of stratospheric ternary solutions and ice clouds. In section 4 we present and discuss the results of the comparisons. Finally, in section 5 we present the conclusions.

## 2 Datasets and methodology

The Cloud-Aerosol Lidar with Orthogonal Polarization (CALIOP) dual-wavelength elastic backscatter lidar (Winker et al., 2009) flies on the Cloud-Aerosol Lidar and Infrared Pathfinder Satellite Observations (CALIPSO) satellite launched in April 2006. The Microwave Limb Sounder (MLS) is onboard the Aura spacecraft launched in July 2004. CALIPSO and Aura are part of the NASA/ESA afternoon "A-train" satellite constellation at 705 km nominal altitude and 98° inclination, with daily near-global coverage attained in 14.5 orbits. The initial A-train configuration of the CALIPSO and Aura spacecraft from April 2006 to April 2008 resulted in an across-track orbit offset of ∼200 km, with the MLS tangent point leading the CALIOP nadir view by about 7.5 minutes. Since April 2008, Aura and CALIPSO have been operated to maintain positioning within tightly constrained control boxes, such that the MLS tangent point and the CALIOP nadir view are colocated to better than about 10-20 km and about 30 seconds.

### 2.1 Reanalysis temperature data

The reanalysis data used are from four analysis centers (NASA, NOAA/NCEP, ECMWF, JMA). Details pertaining to the S-RIP intercomparisons can be found in Fujiwara et al. (2017) and are summarized below.

– **GEOS-5.9.1 :** 0.625° by 0.5° : 3hr : Rienecker et al. (2011) : near real time assimilation system : Goddard Earth Observing System Model, Version 5 : NASA Global Modeling and Assimilation Office (GMAO)

– **MERRA :** 0.666° by 0.5° : 6hr : Rienecker et al. (2011) : Modern-Era Retrospective analysis for Research and Applications : NASA GMAO

– **MERRA-2 :** 0.626° by 0.5° : 3hr : Bosilovich et al. (2015) : Modern-Era Retrospective analysis for Research and Applications Version 2 : NASA GMAO

– **JRA-55 :** 0.582° by 0.56° : 6hr : Kobayashi et al. (2015) : Japanese 55-year Reanalysis : Japanese Meteorrological Agency (JMA)

– **ERA-Interim :** 0.75° by 0.75° : 6hr : Dee et al. (2011) : European Centre for Medium-range Weather Forecasts (ECMWF) interim reanalysis : ECMWF

– **NCEP-CFSR :** 0.5° by 0.5° : 6hr : Saha et al. (2010) : National Oceanic and Atmospheric Administration (NOAA) National Centers for Environmental Prediction (NCEP) Climate Forecast System Reanalysis : NCEP



- All the above except MERRA include GPS RO data during the time period studied here

- GEOS-5.9.1 is a near real-time assimilation product provided to the NASA Earth Observing System (EOS) instrument science teams

The synoptic gridded reanalysis temperatures are interpolated to a common vertical pressure grid (100–10 hPa), with 6 levels per decade ($p = 100 \times 10^{-i/6}; i = 0, 6$), and to the MLS measurement times and geophysical locations.

## 2.2  CALIOP PSC data

We use the CALIOP Level 2 operational dataset L2PSCMask (v1 Polar Stratospheric Cloud Mask Product) produced by the CALIPSO science team. The Level-2 operational data consist of nighttime only data and contain profiles of PSC presence, composition, optical properties, and meteorological information along the CALIPSO orbit tracks at 5 km horizontal by 180 m vertical resolution. We have applied post-processing to generate coarser horizontal/vertical bins for a better comparison at the scale of the MLS along-track and vertical resolution. Each averaging bin is the size of the MLS along-track vertical profile separation (165 km) and the height between the mid-points of the retrieval pressure levels (2.16 km) for the MLS $HNO_3$ data product. This we refer to as the MLS geometric field of view. There are approximately four hundred CALIOP 5 km x 0.18 km "pixels" within the MLS geometric field of view.

## 2.3  MLS gas-phase constituents and temperature

The Microwave Limb Sounder measures thermal emission at millimeter and sub-millimeter wavelengths from the Earth's limb (Waters et al., 2006) along the forward direction of the Aura spacecraft flight track, with a vertical scan from the surface to 90 km every 24.7 s. Each orbit consists of 240 scans spaced at $1.5°$ (165 km) along-track, with a total of almost 3500 profiles per day and latitudinal coverage from $82°$ S to $82°$ N. The Level-1 limb radiance measurements are inverted using 2-D optimal estimation (Livesey et al., 2006) to produce Level-2 profiles of atmospheric temperature and composition. Validation of a previous version of the MLS $H_2O$ and $HNO_3$ data products and error estimations are discussed in detail by Read et al. (2007), Lambert et al. (2007) and Santee et al. (2007). MLS temperature validation and error analysis is discussed by Schwartz et al. (2008). Here we use the MLS version 4 (v4) data (Livesey et al., 2017), which have single-profile precisions (accuracies) of 4–15 % (4–7 %) for $H_2O$, 0.6 ppbv (1–2 ppbv) for $HNO_3$ and for temperature a precision of 0.7 K and a bias in the range $-2$–0 K. We note that MERRA-2 assimilates MLS temperatures, but only at pressures less than 5 hPa and not within the pressure range investigated here (Bosilovich et al., 2015).

Errors in the MLS $H_2O$ contribute a few tenths of a kelvin to the error in frost point temperature and are substantially smaller than the errors in the temperature limb sounding retrievals obtained from MLS. From August 2004 until December 2013, mean differences between NOAA frost point hygrometer and MLS $H_2O$ (Hurst et al., 2014) showed no statistically significant differences (agreement to better than <1%) from 68–26 hPa, although significant biases at 100 and 83 hPa were found to be 10% and 2%, respectively. Increasing the time frame to mid-2015 (Hurst et al., 2016) suggests a long-term drift



in MLS $H_2O$ of up to 1.5% per year starting around 2010. This is still under investigation by the MLS science team, but the effect on the calculated STS reference and frost point temperatures would be less than 0.1 K per year.

## 2.4  COSMIC GPS RO temperatures

We use the US/Taiwan Constellation Observing System for Meteorology, Ionosphere and Climate (COSMIC) network data obtained from the Universities for Cooperative Atmospheric Research (UCAR) COSMIC Data Analysis and Archive Center (CDAAC). GPS RO data have provided high accuracy ($< 0.2$ K, Gobiet et al., 2007), global (day and night) coverage, coupled with excellent long term stability for nearly two decades (Anthes, 2011). The introduction of GPS RO has been documented to improve NWP forecast skill in the ECMWF Integrated Forecast System (IFS) (Bonavita, 2014) and to reduce tropopause and lower stratospheric temperature biases in ERA-I (Poli et al., 2010). The direct assimilation of bending angles or refractivity is now the common practice for many global reanalyses; however, for many other purposes the production of vertical atmospheric temperature profiles from GPS RO data is required. Even though the RO measurements *per se* are SI-traceable (Anthes, 2011), in common with all satellite limb sounding techniques, the retrieval of vertical atmospheric geophysical profiles from RO requires a number of assumptions because of the long ray path through a non-uniform atmosphere (Ho et al., 2012). Therefore, corrections are required for ionospheric effects, variations in water vapor, and gradients in temperature along the ray path (Poli and Joiner, 2004; Anthes, 2011). Many other studies have intercompared GPS RO with independent operational analyses, e.g. with forecast versions that have not assimilated the GPS RO data. The near real time COSMIC data (in the form of bending angles or refractivity) are ingested by most of the data assimilation procedures considered here (except for MERRA), and therefore these reanalyses are not strictly independent of the postprocessed COSMIC temperatures. We have chosen to use the COSMIC temperatures as a common reference to evaluate the reanalysis departures, rather than using the reanalysis ensemble mean.

Staten and Reichler (2008) examined the tropopause temperatures obtained from various GPS RO datasets and determined a global mean bias of $<0.1$ K between the different RO instruments and of $<0.5$ K between RO instruments and radiosondes. A later study by the same authors (Staten and Reichler, 2009) reported a more comprehensive analysis of the RO temperature precision for the COSMIC data that examined variations with height, latitude and season. The concept of apparent precision was introduced, whereby the temperature differences between colocated neighboring RO measurements are organized by their temporal and spatial differences. A quadratic polynomial surface was fit to the RMS temperature differences, acting as a noise filter for the atmospheric variability and providing the apparent temperature precision of perfectly colocated data via extrapolation of the fitted RMS values to the time and distance origin. For the 15 km to 26 km height range, the RMS differences increase monotonically from about 0.5 K to 1.5 K. The results presented graphically by Staten and Reichler (2009) appear to be applicable to the RMS of colocated measurement pairs and should therefore be divided by $\sqrt{2}$ to represent the precision on a single GPS measurement.

Alexander et al. (2014) estimated the COSMIC GPS RO temperature precision by examining the RMS difference between a large ensemble of pairs of independent COSMIC observations (both wet and dry temperature profiles). Temperature accuracy was assessed by comparing to colocated ECMWF analyses, and similar results were obtained over data sparse (Pacific) and





data dense (USA) regions, with an average bias of 0.1 K (RO−ECMWF) for 10 to 30 km and −0.5 K above 35 km. Comparisons of COSMIC GPS RO with ERA-I (Dee et al., 2011) showed deviations of ±1 K at 100 hPa and ±2 K at 10 hPa for the polar regions. Ladstädter et al. (2015) made COSMIC GPS RO comparisons to Vaisala RS90/92 radiosondes and found mean temperature differences for 100–30 hPa of <0.2 K since 2006 and of <0.5 K for 30–10 hPa. Scherllin-Pirscher et al.

(2011b) modeled the observed vertical error structure of GPS RO measurements, including the dry temperature error variation. Differences between RO instruments from different missions were found to have global mean standard deviations differing by at most 0.2 K at all levels between 4 and 35 km. The GPS RO model error, $s_{model}$, is a constant value, $s_0 = 0.7$ K around the tropopause, increasing exponentially with height into the stratosphere and given by

$$s_{model} = \begin{cases} s_0 & \text{for } z_{Ttop} < z < z_{S_{bot}}, \\ s_0 \, exp\left[\frac{z-z_{S_{bot}}}{H_{S_{bot}}}\right] & \text{for } z_{Sbot} \leq z \end{cases} \qquad (1)$$

where $z_{Ttop}$ is the top level of the tropopause, $z_{Sbot}$ is the bottom level of the stratopause. Latitudinal and seasonal variations are governed by an adjustable scale height parameter, $H_S$, which has its lowest values at high latitudes and during the winter months. The GPS model error has a different functional form for the troposphere (Scherllin-Pirscher et al., 2011b), which we do not require here.

The GPS RO data provide a measurement of the atmospheric refractivity, $N$, which is dependent on the temperature, $T$ (in

K), atmospheric pressure, $p$ (in hPa), and the water vapor partial pressure, $p_w$ (in hPa). For heights above 4 km, and following correction for ionospheric electron density, $N$, is given by (Smith and Weintraub, 1953)

$$N = c_1 \frac{p}{T} + c_2 \frac{p_w}{T^2} \qquad (2)$$

where $c_1 = 77.60$ K hPa$^{-1}$ and $c_2 = 3.73 \times 10^5$ K$^2$ hPa$^{-1}$. The first term in the equation represents the contribution to refractivity from the total molecular polarizability of the dry atmosphere, whereas the second term corresponds to the effects of

the permanent dipole moment of water vapor molecules. In regions of low atmospheric humidity, above 14 km at low latitudes and above 9 km at high latitudes, dry temperatures provide an adequate representation of the physical atmosphere (Scherllin-Pirscher et al., 2011a). We used the latest available COSMIC GPS version 2013.3520 "atmPrf" dry temperatures processed at the UCAR CDAAC. These have been compared recently to Vaisala RS92 radiosonde temperatures (Ho et al., 2017), and the mean radiosonde − RO global daytime temperature difference over 200–20 hPa was found to be 0.20 K, with a mean standard

deviation of 1.5 K. For completeness, we also evaluated the "wetPrf" wet temperatures for the same data version from 2008 to 2013 in the polar regions. We obtained mean temperature differences (dry − wet) in the range −0.1 to +0.05 K at 100 hPa and −0.4 to +0.15 K at 10 hPa, with standard deviations of 0.15 K at 100 hPa increasing to 1.1 K at 10 hPa. The wet temperatures are not considered further in this paper.

We have not implemented the line of sight corrections (Feltz et al., 2014b, a) that are possible when comparing against

gridded temperature fields. On a case-by-case basis the correction can be >1 K, depending on the orientation of the GPS raypath with respect to temperature gradients, but the raypath averaging method has been shown to reduce the bias (Feltz et al., 2014b, a). For larger sample sizes the biases are reduced because the positive/negative temperature gradients are averaged away even for a simple closest matching pair method (Feltz et al., 2014b, a).



For convenience we interpolated (in log pressure) the COSMIC temperature profiles to a set of fixed pressure levels with 24 levels per decade ($p = 1000 \times 10^{-i/24}; i = 0, 49$, approximately 2/3 km vertical resolution). In Section 4.3 we compare each reanalysis data set to the COSMIC data and calculate the bias and standard deviation of the temperature differences.

## 3  Thermodynamics of PSC formation

The growth of solid nitric acid trihydrate (NAT) crystals is kinetically limited (e.g. Voigt et al., 2005) and is frequently out of equilibrium with the gas-phase $HNO_3$ abundance. This makes NAT PSCs unsuitable for use as a local temperature reference, since their growth by $HNO_3$ uptake is dependent on the temperature history (Larsen et al., 1997). On the other hand, liquid STS reacts more quickly to ambient temperature changes, effectively acting as a thermometer (Hoyle et al., 2013), provided that very rapid temperature changes are excluded (Voigt et al., 2005). Ice PSCs also react quickly to temperature changes, because of

the high ambient $H_2O$. Critical supercooling is the temperature depression below the frost point where homogeneous freezing of the STS can take place to form ice (Larsen, 2000). A depression of about 3 K is typically required (Carslaw et al., 1998; Koop et al., 1998). However, heterogeneous nucleation, such as the formation of ice on pre-existing NAT or freezing of STS mediated by active sites on foreign nuclei immersed within the liquid drop (Engel et al., 2013), does not require a temperature depression below the frost point.

Theoretical existence temperatures of the PSC types are calculated using equilibrium thermodynamics and are dependent on the ambient partial pressures of $H_2O$ in the case of the ice frost point, $T_{ice}$ (Murphy and Koop, 2005), and also $HNO_3$ for STS (Carslaw et al., 1995). Errors in the calculations of these reference temperatures arising from uncertainties in the MLS $H_2O$ and $HNO_3$ data are estimated to be $\leq 0.5$ K for $T_{ice}$ and $\leq 0.7$ K for $T_{eq}$ in the pressure range 68–21 hPa.

It is expected that theoretical equilibrium calculations are more appropriate for 'steady state' conditions, e.g. PSCs produced

by slow synoptic cooling, than rapidly changing temperatures associated with the cold phases of mountain waves (Dörnbrack et al., 2001). Therefore, mountain wave production of ice that is detected by the CALIOP wave-ice classification is excluded from this study. However, mountain wave activity also influences the production of STS (Carslaw et al., 1998). This can be potentially excluded to some extent by analysis of the wave activity in the reanalysis datasets (Knudsen, 2003) and rejection of data in the affected locations. Alternatively, since the CALIOP data have resolution superior to that of the reanalysis grids, we

can address the challenge posed by the disparity between the CALIOP, MLS, and reanalysis horizontal and vertical sampling scales by selecting only those scenes with a consistent CALIOP PSC classification that fills a substantial proportion of the synoptic-scale MLS field of view (FOV) (see Section 4.5). Therefore, we can effectively mitigate the visual speckle effect (Pitts et al., 2009) of disparate PSCs occurring within the MLS field of view generated by sub-grid temperature fluctuations that are not resolved by the reanalysis data or that arise because of random noise in the CALIOP signals. Scenes satisfying this

requirement for CALIOP STS detections we denote as `LIQ`, and for CALIOP ice detections we denote as `ICE`.



### 3.1 Ice cloud equilibrium

The frost point temperature (FPT, $T_{ice}$) variation over the lower stratosphere is shown in Figure 1(a) for 5 ppmv $H_2O$. The FPT derivative, shown in Figure 1(b), indicates that a change in $H_2O$ of 1 ppmv results in a change in FPT of ∼1.05–1.2 K. The FPT derivative is used to convert the estimated MLS $H_2O$ single-profile retrieval errors (Livesey et al., 2017) into an equivalent error in FPT and the result is shown in Figure 1(c). The accuracy and precision are estimated to be 0.6–0.3 K and 0.3 K from 68–21 hPa, respectively. The root mean square error (RMS) is the quadrature addition of the accuracy and precision. Hence the estimated RMS errors in FPT arising from the uncertainties in the MLS $H_2O$ measurements from 68–21 hPa are 0.7–0.4 K. As noted this is the error associated with a single measurement; although the statistical precision can be assumed to diminish as the square root of the number of measurements used for averaging, the accuracy remains as the overriding error source. In any case, the error in FPT is substantially lower than the estimated RMS errors for the MLS lower stratosphere temperature measurements (Section 2.3).

### 3.2 Supercooled ternary solution (STS) equilibrium

Sample STS equilibrium curves are calculated using the analytic formula of Carslaw et al. (1995) and shown in Figure 2(a) for a range of lower stratospheric conditions; pressures 68, 46, and 32 hPa; $H_2O$ mixing ratios 4, 5, and 6 ppmv; and $H_2SO_4$ mixing ratios 0.1 and 0.5 ppbv. Figure 2(b) shows the effect of removing the water vapor partial pressure variation through a coordinate transformation of the temperature scale relative to $T_{ice}$. In Figure 2(c) we show that the estimated accuracy and precision in $T_{eq}$ arising from the uncertainties in the MLS $HNO_3$ measurements from 68–21 hPa are 0.6–1.5 K and 0.4 K, respectively. The corresponding RMS errors are 0.7–1.6 K. This is greater than the FPT error, but still lower than the estimated RMS errors for the MLS lower stratosphere temperature measurements (Section 2.3).

## 4  Results

### 4.1 Temperature fluctuations

Mesoscale temperature fluctuations have a significant effect on the formation of PSCs (Murphy and Gary, 1995; Dörnbrack and Leutbecher, 2001; Gary, 2006), and therefore the capability of models to accurately mimic PSC formation requires consideration of these sub-grid processes (Engel et al., 2014; Hoyle et al., 2013). The high vertical resolution of the COSMIC temperatures allows an examination of the spectrum of the temperature variance over the height region of PSC formation. We have analysed the atmospheric temperature variability in the 80 to 20 hPa range by fitting cubic polynomials to the COSMIC profiles and calculating the mean variance, $\overline{T'^2}$, over this range. The cubic fit provides data smoothing to estimate the unperturbed background temperature state (Whiteway, 1999; Knudsen, 2003), and therefore the mean variance is a statistic that serves as a simple measure of the vertical wave disturbances. The orientation of the COSMIC line of sight to the planes of the gravity wave fronts (Preusse et al., 2002) has an influence on the detected wave amplitudes and hence the variances. No correction for this effect is attempted.





The mean temperature variance relative to the background unperturbed temperature state, i.e. the mean fractional variance $\overline{\left(\frac{T'}{\bar{T}}\right)^2}$, is related to the potential energy density of the wave disturbance, $E_p$, by the relation (Whiteway, 1999; Tsuda et al., 2000):

$$E_p = \frac{1}{2}\left(\frac{g}{N}\right)^2 \overline{\left(\frac{T'}{\bar{T}}\right)^2} \quad \text{J Kg}^{-1} \tag{3}$$

where $g$ is the gravitational acceleration, $T'$ is the perturbation amplitude, $\bar{T}$ is the mean temperature, and $N$ is the bouyancy frequency. Wave activity was categorized by Whiteway (1999) and Knudsen (2003) as inhibited for $E_p < 1$ J Kg$^{-1}$ and enhanced for $E_p > 2$ J Kg$^{-1}$. Assuming typical values for the polar lower stratosphere, $N \sim 0.02$ s$^{-1}$, $\bar{T} = 200$ K, we find that an energy density of $E_p = 1.5$ J Kg$^{-1}$ corresponds to $\overline{T'^2} = 0.5$ K$^2$, and we have used this value as a convenient variance threshold to delineate the transition between quiescent and enhanced wave activity.

The mean (2008–2013) temperature variances for the southern and northern hemispheres are shown in Figure 3 for temperatures below 200 K. The Arctic temperature variance spectrum in Figure 3(d) shows a longer tail, extending well beyond 1 K$^2$, than for the Antarctic in Figure 3(a). The geographic distributions for the profiles with variances exceeding the 0.5 K$^2$ threshold are shown in Figures 3(b,e). The signature of orographically forced gravity waves over the Antarctic Peninsula is captured by the variance statistic, even though it is not specifically tuned to detect orographic waves. Similarly, in the Arctic

a slight increase in variance can be seen along the East coast of Greenland. These spatial imprints of gravity wave activity are largely consistent with the patterns identified by the AIRS 15 μm brightness temperature variations (Hoffmann et al., 2017). However, since any kind of wave activity giving rise to vertical temperature fluctuations will increase the calculated variance, non-orographic features are also visible in these temperature variance maps. Figures 3(a) and (d) indicate that around 4% and 12% of the COSMIC profiles exceed the 0.5 K$^2$ ($E_p = 1.5$ J Kg$^{-1}$) threshold in the southern and northern hemispheres,

respectively.

The standard deviation of the differences between the reanalysis and COSMIC temperatures is the result of the quadrature addition of their standard deviations. Small-scale temperature fluctuations cannot be captured accurately by the reanalysis data, but the COSMIC data should perform better because of their higher vertical resolution. A decrease in the spread of the temperature differences of reanalysis data compared to COSMIC should be detectable by rejecting the temperature profiles

from consideration that are expected to have lower fidelity to the true atmosphere. This is seen to be the case in Figures 3(c) and (f), where the standard deviations of the temperature differences between ERA-I and COSMIC improve successively as the more highly fluctuating profiles (as determined by the COSMIC mean temperature variances) that are less well captured by the ERA-I reanalysis scheme are excluded. The mean differences are also seen to improve substantially for the Arctic, but not for the Antarctic. Another effect to consider is that this method also reduces the variability of the atmospheric path

along the COSMIC line of sight, and consequently the fidelity of the COSMIC temperatures to the true atmosphere will also be improved. Therefore, we cannot attribute the entirety of the observed reduction in the standard deviation to the ERA-I reanalysis data alone.



In the reanalysis temperature comparisons with COSMIC in Section 4.6 we have excluded profile matches with COSMIC temperature variance $>0.5$ K$^2$ ($E_p >$1.5 J Kg$^{-1}$), and also restricted the COSMIC temperatures to below 200 K to better match the regions of potential PSC formation.

## 4.2 Orographic gravity wave case study

In the next section we report on the statistical analysis of the temperature differences, but first we examine a case study taken from a synoptic cooling period coupled with a gravity wave event on 28 August 2008 (2008d241) over the Antarctic Peninsula region as shown in Figure 4. The 50 hPa maps (Figure 4(a)) show the reanalysis temperatures at 18UT on 28 August 2008, and a large variation in the wave activity is seen, with JRA-55, ERA-I and NCEP having much larger amplitudes than GEOS-5.9.1, MERRA and MERRA-2. A longitude section from $-70°$ to $-50°$ at latitude 72° S is selected for further study. The mountain

wave is forced by flow over the topographic ridge along the Antarctic peninsula, and the wind direction is almost orthogonal to the ridge line, with wind speeds $u \sim$ 25–35 ms$^{-1}$. The vertical wavelength, $\lambda_v$, is about 7–8 km, with a temperature amplitude of $\pm12$ K determined by MLS. PSC ice formation detected by CALIOP (not shown) correlates with the cold phases of the gravity wave, but not with all regions colder than the frost point. The horizontal wavelength, $\lambda_h \sim$ 400–500 km, is resolved by AIRS (not shown) using the techniques of Eckermann et al. (2009). From the linear gravity wave dispersion relation we obtain

the vertical wavelength, $\lambda_z = 2\pi u/N$, in the range 7-11 km, in agreement with that observed.

Figure 4b shows the temperature variation at five pressure levels (100, 70, 50, 30 and 20 hPa) along the longitude section. Temperatures below the ice frost point, calculated using MLS H$_2$O, are highlighted by square symbols, and the mountain terrain is indicated by the black shading. At 50 hPa the peak to trough temperature differences are 7.4, 6.2. 8.0, 10.0, 6.1 and 14.9 K for GEOS-5.9.1, MERRA, MERRA-2, JRA-55, ERA-I and NCEP-CFSR, respectively; i.e., over a factor of two difference is

seen in the temperature amplitudes between the reanalyses. Figures 4(c) and (d) show the time variation of the gravity wave induced temperature disturbance during 28-29 August, along the longitude section at 50 hPa and 30 hPa, respectively. It is clear from these figures that the height variation, timing of the onset and extent of temperatures cold enough to potentially form ice PSCs are quite different in each of the reanalyses. The lowest temperatures are below the ice frost point at every time stamp for the 30-hPa NCEP-CFSR data, whereas for ERA-I the temperatures are only sporadically below the ice frost point. In

this particular case, a PSC model run based on the NCEP-CFSR temperatures would clearly produce much larger, and longer lasting, ice volumes than one using the ERA-I temperatures.

## 4.3 Reanalysis temperature difference distributions compared to COSMIC, $\mathbf{T_{re} - T_{ro}}$

Distributions of the temperature differences between the reanalyses and COSMIC were compiled for the polar regions for the 100–10 hPa pressure range. The geophysical structure of the temperature differences has been obtained from summary

statistics consisting of zonal medians and standard deviations calculated over the 2008–2013 period (Figure 5). Both JRA-55 and NCEP-CFSR show abrupt vertical transitions in the median differences above 21 hPa in both hemispheres. This is likely to be an interpolation error in the case of NCEP-CFSR since the 14.7 hPa common vertical grid level is distant from the closest native vertical grid points at 20 and 10 hPa. However, for JRA-55 the native vertical grid has a higher vertical resolution and




the interpolation error is small. The standard deviations for the Arctic are similar in zonal structure for all the reanalyses, but the values are systematically smaller in the Antarctic.

Figure 6 is a summary of the temperature bias ranges, extracted from the data shown in Figure 5, that readily captures the salient information over the pressure range from 68 to 21 hPa, which encompasses the bulk of the PSC vertical formation

region (Pitts et al., 2009). The main attributes of the reanalyses are revealed by the relative size of the boxes and their location relative to the zero line. The box width is set to the largest difference in minimum to maximum bias at any height (68–21 hPa), and the box height is set to the largest difference in minimum to maximum meridional bias. The MERRA boxes are larger than those for any of the other reanalyses, with widths (heights) of 0.54 K (0.54 K) in the Antarctic and 0.86 K (0.50 K) in the Arctic. The smallest box widths are for Arctic NCEP-CFSR (0.26 K) and Antarctic JRA-55 (0.26 K). The smallest box heights

are for Arctic NCEP-CFSR (0.18 K) and Antarctic MERRA-2 (0.14 K).

For both polar regions the reanalysis temperature differences from COSMIC are encompassed within a quite narrow range of $-0.61$ to $+0.48$ K. GEOS-5.9.1, MERRA, MERRA-2 and JRA-55 have cold biases ($-0.61$ to $-0.05$ K) in the Antarctic, whereas ERA-I has a warm bias ($+0.04$ to $+0.38$ K). The NCEP-CFSR bias ($-0.30$ to $+0.16$ K) changes with decreasing pressure from warm to cold. Overall, the average Arctic temperature biases for individual reanalyses with respect to COSMIC

are shifted by $\sim$0.1–0.4 K from the Antarctic values.

### 4.4 Relative proportion of CALIOP PSC types within MLS geometric FOV

Figure 7 is an example of the probability density distribution, presented as a ternary graph, of the relative proportion of the three groups of CALIOP PSC types, STS, MIX and ICE occurring within the MLS geometric field of view. Here we define the congener names MIX, for the sum of the MIX1, MIX2, and MIX2-enh types, and ICE, which also includes wave-ice. This

figure is compiled by accumulating the co-existence frequency of these three groups of PSCs for the 2013 Antarctic winter on the 46 hPa pressure level for temperatures below the ice frost point (as determined by ERA-I). The three axes lie along the sides of an equilateral triangle, and the three components are constrained to sum to unity. At the center of any side the percentage of the two adjoining PSC types is 50:50, and the third type at the opposite vertex is zero. At the center of the triangle the proportion of all three PSC types is equal to 1/3. The most populated areas consist of two branches close to the MIX/STS and

MIX/ICE axes. Away from these two branches, the STS/ICE region with less than 10% MIX is essentially devoid of data. This graph is a striking demonstration that the co-existence of STS and ice PSCs on a spatial scale of the size of the MLS geometric field of view is relatively rare. The implication is that the rapidity of freezing of STS to form ice and the resulting re-equilibration of the water vapor and cloud ice is faster than can be captured by the satellite instrument integration times. The A-train orbital speed is $\sim$7.5 km s$^{-1}$, and hence the 165 km along-track MLS FOV is traversed in about 22 s.

### 4.5 MLS field of view PSC fill-fraction

We illustrate the concept of the field of view fill-fraction in Figures 8-10. Figure 8 shows CALIOP and MLS data recorded over a 7000 km transect over Antarctica on 23 June 2008 (2008d175). The CALIOP PSC types show large contiguous areas of STS and ice, but with other embedded PSC types causing visual speckle as mentioned above (Figure 8(a)). The MLS





temperatures are up to 4 K below the frost point (Figure 8(b)). Large regions of $HNO_3$ depletion are seen where abundances have been reduced to practically zero values by sequestration and denitrification by PSCs compared to background values of over 12 ppbv (Figure 8(c)). A smaller dehydrated region has $H_2O$ values 1-2 ppmv lower than the ~5 ppmv background (Figure 8(d)). The second and third rows in the figure show expanded views of a selected region at 32 hPa centered on a

particular MLS geometric field of view. At each magnification step the CALIOP data reveal increasingly finer detail down to the 5-km by 0.180 km pixel size, whereas the MLS data become over-sampled.

Figure 9 shows the corresponding CALIOP PSC types and the Feature Mask information within the MLS geometric field of view shown in Figure 8. The CALIOP feature mask is a 3-digit number, `N1N2N3`, where `N1` encodes information on the height of each "pixel" relative to the local tropopause (Figure 9(b)) and `N2N3` encodes information on the horizontal averaging (5 to

135 km) and the backscatter threshold used for detection (Figure 9(c)). In this particular case, all pixels are at least 4 km above the tropopause, and the majority of pixels are detected with a horizontal averaging of 45 km or smaller.

The fraction of the MLS FOV occupied by the different PSC types (Figure 9(a)) is calculated and given in the figure caption. We impose an upper limit on the occupancy fraction of cloud-free pixels of 0.25 and require the fraction of the dominant `LIQ` or `ICE` PSCs to be greater than 0.75. The example scene has too great a variation of PSC types within the MLS field of view

and fails this acceptance test since it only has a 0.69 `LIQ` fill-fraction. Orbit transects of the MLS FOV PSC fill-fraction for the different PSC types are shown in Figure 10. The `LIQ` and `ICE` filled FOV threshold criteria are mutually exclusive. Despite the PSCs extending over a few thousand km, the strict acceptance criteria, albeit essential to ensure a more uniform single PSC type across the MLS FOV, severely reduces the total number of scenes available for analysis.

## 4.6 Reanalysis temperatures compared to `LIQ` and `ICE` reference points

We select viewing scenes in which there is a distinct dominant PSC classification in a sample volume that is similar in size to the MLS gas species resolution. As discussed above the requirement is that 75% or more of the CALIOP pixels in the MLS geometric FOV have the same PSC classification.

We summarize the method below:

– Identify `LIQ` and `ICE` PSCs using the CALIOP lidar measurements.

– Accumulate the CALIOP PSC types (`LIQ` and `ICE`) at the MLS along-track resolution (165 km × 2.16 km), ensuring that the same PSC type is detected in at least 75% of the MLS field of view.

– Calculate the theoretical temperature dependence of STS ($T_{eq}$) and ice ($T_{ice}$) PSCs under equilibrium conditions using the spatially and temporally colocated MLS gas-phase $HNO_3$ and $H_2O$ measurements.

– Compare (a) calculated and observed $HNO_3$ uptake in STS and (b) ice temperature distribution vs frost point with

reanalysis data and MLS temperatures.

– Create `LIQ` and `ICE` temperature distributions for each data set: GEOS-5.9.1, MERRA, MERRA-2, JRA-55, ERA-I, NCEP-CFSR (all sampled by interpolating to the MLS measurement times and locations) and MLS temperature.



    – Calculate median and mean temperature deviations from $T_{eq}$ and $T_{ice}$ and their standard deviations for LIQ and ICE classifications, respectively.

## 4.7 Distributions of observed MLS and theoretical equilibrium $HNO_3$ and $H_2O$ gas-phase variation with temperature

Figure 11 shows scatter plots of the coincident MLS $HNO_3$ and $H_2O$ for Antarctic PSCs classified by CALIOP vs ERA-I temperature at 46 hPa in 2013. Figures 11(a) and (c) show the scatter of $HNO_3$ vs temperature and inverse temperature, respectively, along with the theoretical $HNO_3$ gas-phase uptake curves for STS and NAT, indicating clearly that LIQ PSCs are closely associated with the STS curve. Similarly, Figures 11(b) and (d) show the scatter of gas-phase $H_2O$ and its close association with the frost point temperature in the presence of ICE PSCs.

In Figure 12 we show the temperature variation of the estimated partial pressure of water vapor over ice, $p_{ice}$, for atmospheric pressure levels in the range 100–10 hPa. The partial pressure is determined simply from the product of the MLS $H_2O$ volume mixing ratio and the atmospheric pressure. Since CALIOP indicates the presence of coincident ice PSCs, this is an estimate of the partial pressure of water vapor over ice. The bias in the MLS temperatures is evident in the departure of the $p_{ice}$ observations from the theoretical equilibrium curve in Figure 12(a), whereas the ERA-I temperatures place the $p_{ice}$ observations in much

better agreement.

## 4.8 Polar temperature reference points

   In Figure 13 we show the variation of the MLS gas-phase $HNO_3$ with ERA-I temperature corresponding to CALIOP PSC classifications at 32 hPa. The MLS $HNO_3$ data are separated into corresponding CALIOP PSC categories, allowing comparison of observed and modeled uptake of $HNO_3$ in different types of PSCs. The scatter of MLS $HNO_3$ against the temperature

deviation from the frost point (calculated using MLS $H_2O$) is shown in Figure 13(a) for LIQ and ICE PSCs. The $HNO_3$ gas-phase uptake in the presence of liquid-phase LIQ PSCs follows the STS equilibrium curve. In contrast, the $HNO_3$ abundance is very low in the presence of ICE PSCs. In Figure 13(b), the $HNO_3$ gas-phase uptake in clouds associated with the CALIOP solid NAT type MIX2 shows significant non-equilibrium variation and lies between the STS and NAT equilibrium curves. Histograms of the temperature distributions of (a) and (b) are shown in Figure 13(c). The light blue (LIQ) and yellow (MIX2)

colored histograms are for $HNO_3$ mixing ratios >1 ppbv, whereas the red (LIQ) colored histogram is for $HNO_3$ mixing ratios <1 ppbv. The tails of the temperature distributions for LIQ (light blue and red histograms) do not extend to temperatures lower than those in the distribution for ICE (dark blue), and no peaks are observed that would indicate the existence of PSCs at a frost-point depression near $T_{ice} - 3$ K. In Figure 13(d), the temperatures are transformed according to the STS equilibrium curve for the LIQ classification and NAT equilibrium curve for the MIX2 classification; the ICE classification remains the

same as in (c) for comparison. Note that the LIQ histogram is shifted and becomes much narrower (Lambert et al., 2012). This is just an illustrative case; since there are seven temperature data sources, collected over six years on six pressure levels, the total number of histograms is 252.



### 4.8.1 Statistics of the temperature difference distributions, $T_{re}-T_{eq}$, for each reanalysis and comparison to the normal distribution

Statistics for each reanalysis data set of the temperature difference distributions were generated from histograms analogous to those in Figure 13(d) for each year and pressure level. The combined data for the 2008–2013 Antarctic winters at 46 hPa

are displayed in Figure 14 for the `LIQ` PSCs and Figure 15 for the `ICE` PSCs. The distributions for `LIQ` PSCs all show negative temperature biases compared to the STS equilibrium reference, with mean values in the range $-0.9$ to $-0.3$ K and standard deviations $\sim$0.6 K. The standard error of the mean is $\sim$0.01 K, and the medians are very similar to the mean values, indicating a small skewness of the distribution. Visually, the scatter follows a normal distribution reasonably well in the tails of the distribution, but with a mismatch near the peak of the distribution. The normalized $\chi^2$ values are in the range 1.4–2.2,

which for around 30 degrees of freedom (i.e. the difference between the number of histogram bins and the number of fitted parameters) indicates a poor goodness-of-fit. We note that improved fits can be obtained by introducing a secondary normal distribution with a smaller standard deviation (not shown), which seems to imply the underlying presence of two distinct modes of temperature precision in the reanalyses. We have not investigated this finding any further, but one pertinent avenue of exploration, given that the density of observations ingested into the system is sparse in the polar regions, would be to look

for evidence that the reanalysis temperatures near grid points that are located closer to the geographic source of assimilated observations have better precision than those at grid points remote from observations.

    The distributions for `ICE` PSCs show mainly negative temperature biases compared to the equilibrium ice frost point reference. Mean temperature bias values for `ICE` PSCs are in the range $-0.6$ to $0$ K with standard deviations of $\sim$0.7 K, both of which are larger than their `LIQ` PSC counterparts. The `ICE` distribution fits were also improved in some cases by introducing

a secondary normal distribution.

### 4.8.2 Reanalysis temperature difference distributions compared to the `LIQ` and `ICE` references, $T_{re}-T_{eq}$, and COSMIC GPS RO, $T_{re}-T_{ro}$

Intercomparisons of the reanalysis temperature distributions derived from the data in Figures 14 and 15 are displayed in Figure 16 for the Antarctic in 2013 (a representative year). In addition to the differences relative to the `LIQ` and `ICE` reference

points, we also show the comparison with the COSMIC temperatures (latitude range 90° S to 60° S, mean variance $<$0.5 K$^2$ and COSMIC temperatures $<$200 K). The largest bias is for MLS, and biases for the `LIQ` reference are consistently about 0.5 K more negative than for `ICE`. Biases for the COSMIC reference are smaller than those for `LIQ` or `ICE`. MERRA does not assimilate the COSMIC GPS RO data, and it has the largest bias with respect to the COSMIC reference. Standard deviations for `LIQ` are consistently smaller than those for `ICE`, and standard deviations for COSMIC are between those of `LIQ` and `ICE`.

Interannual variability for the 46 hPa level is shown in Figure 17, where median values for the COSMIC reference display less variability than those for the `LIQ` or `ICE` references. Two years (2010 and 2012) stand out in the `LIQ` median reference as anomalously high, and we note that those Antarctic winters were warmer than in the other years (Kuttippurath et al., 2015). Standard deviations show less interannual variability than median values, especially for the COSMIC reference.





The vertical temperature differences over the 100–10 hPa range are shown in Figure 18 for the Antarctic winters 2008–2013. Median values for the `ICE` reference are more constant with height than those for the `LIQ` reference, which become more negative with increasing height. Median values for the `LIQ` reference are biased lower than those for `ICE` by ∼0.5 K, although standard deviations for the `LIQ` reference are smaller than those for `ICE`. The corresponding observations for the

Arctic are shown in Figure 19, but there the number of data points in the `ICE` reference is much lower. Again, MLS shows the largest bias.

Das and Pan (2014) intercompared COSMIC GPS RO temperatures with satellite measurements from the Sounding of the Atmosphere using Broadband Emission Radiometry (SABER) instrument and Aura MLS and determined that the structure of the observed median temperature differences was due in large part to inherent retrieval biases of SABER and MLS (December

2010 to November 2011). Seasonal and meridional bins were used to compare with SABER and MLS. At high latitudes, in the 100–10 hPa range, median profiles for SABER showed an obvious positive bias (2–3 K higher than COSMIC) in the lower stratosphere, increasing with decreasing atmospheric pressure. Median profiles for MLS showed negative biases of 0 to 2 K, but with larger oscillations than seen with SABER. These results indicate the difficulty of obtaining sub-kelvin temperature accuracy with microwave and infrared limb sounders.

The standard deviations result from the combination of the quadrature addition of the precisions of the reanalyses and the reference points. In Figures 1 and 2, we estimated the minimum precision of the `ICE` and `LIQ` references to be ∼0.3 K. For the `LIQ` reference comparison (Figures 18(d) and 19(d)), the minimum combined precision is 0.5 K (in both NH and SH); therefore, subtracting the `LIQ` reference precision results in an estimated precision of 0.4 K for the reanalysis data. The theoretical vertical structure of the COSMIC GPS RO standard deviation values follows an exponential increase with height

as shown by Scherllin-Pirscher et al. (2011b) and is defined by Equation 1. We have determined the best fit parameters for each hemisphere from Equation 1 by minimizing the RMS deviations of the ensemble means of the (reanalysis − COSMIC) standard deviations. The solid black lines in Figures 18(f) and 19(f) are the quadrature addition of the estimated reanalysis temperature error, $\sigma_{T_{RE}} = 0.4$ K, and the GPS RO model error given in Equation 1 with best fit parameters $s_0 = (0.83, 0.69)$ K, $H_s = (10.1, 8.8)$ km, and $Z_S = (22.5, 23.1)$ km for the Arctic and Antarctic, respectively; i.e. our values for $s_0$ are close to the

0.7 K value given by Scherllin-Pirscher et al. (2011b).

The MERRA temperatures should be uncorrelated with the COSMIC data, and in the main they do show a higher combined standard deviation than the other reanalyses. Some correlation with the other reanalyses that assimilate COSMIC data may be expected to reduce the standard deviation, and this appears to be the case except for the top two pressure levels.

For the southern hemisphere, the `LIQ` reanalysis standard deviation (Figure 18(d)) appears to be approximately constant

from 46 to 14 hPa, rising at the lower stratospheric levels, whereas the northern hemisphere (Figure 19(d)) shows no increase at the lower levels. Overall the combined standard deviations for the reanalyses and COSMIC data are higher in the Arctic (Figure 19(f)) than in the Antarctic (Figure 18(f)) by ∼0.1 K.

Although we have not directly matched A-train locations with the COSMIC occultations, we can estimate the differences between the COSMIC temperatures and the thermodynamic reference temperatures by elimination of the reanalysis temperature

biases as follows. First, we decompose the reanalysis, radio occultation and equilibrium temperatures into $T_x = T' + \Delta T_x$,





where T$'$ is the true temperature, and the temperature bias is $\Delta T_x$, such that $x = re, ro, eq$ for the reanalysis, radio occultation and equilibria, respectively. Then, we have $\overline{(T_{re} - T_{ro})} = \overline{\Delta T_{re}} - \overline{\Delta T_{ro}}$ and $\overline{(T_{re} - T_{eq})} = \overline{\Delta T_{re}} - \overline{\Delta T_{eq}}$. With the tacit implication that the $\overline{\Delta T_{re}}$ are unchanging and can be eliminated from both equations, we derive $\overline{(T_{re} - T_{eq})} - \overline{(T_{re} - T_{ro})} = \overline{\Delta T_{ro}} - \overline{\Delta T_{eq}}$. The results of these operations are shown in Figure 20 for LIQ and COSMIC, $\overline{\Delta T_{LIQ}} - \overline{\Delta T_{COSMIC}}$, and ICE

and COSMIC, $\overline{\Delta T_{ICE}} - \overline{\Delta T_{COSMIC}}$. The temperature bias difference profiles for the different reanalyses are very closely grouped, especially over the pressure range 68–21 hPa, justifying the assumption that $\overline{\Delta T_{re}}$ can be eliminated. The biases for $(\overline{\Delta T_{LIQ}} - \overline{\Delta T_{COSMIC}})$ are similar in their shape and value for both hemispheres over the pressure range 68–21 hPa, with $\overline{\Delta T_{LIQ}}$ around 0.5 to 1.0 K lower than $\overline{\Delta T_{COSMIC}}$. The biases for $\overline{\Delta T_{ICE}}$ are around 0 to 0.5 K lower than $\overline{\Delta T_{COSMIC}}$ for the Antarctic, but the Arctic has too few data points to make a meaningful comparison. The LIQ and ICE profiles for the

Antarctic have similar values at 32 hPa with $\overline{\Delta T_{ICE}} - \overline{\Delta T_{COSMIC}} \sim -0.5$ K, whereas the LIQ values diverge from those of ICE by up to $-1.5$ K above and below this level. The reasons for these LIQ and ICE systematic discrepancies are not known; however, the differences are within the expected uncertainty limits in the 68–21 hPa pressure range.

Finally, a summary of the mean temperature bias ranges of the reanalyses relative to the LIQ ($-1.6$ to $-0.3$ K) and ICE ($-0.9$ to $+0.1$ K) equilibrium references and COSMIC ($-0.5$ to $+0.2$ K) is shown in Figure 21. The ranges quoted are for

the pressure range 68–21 hPa. For all reference points, the coldest reanalysis biases tend to be found in the Antarctic and the warmest in the Arctic.

## 5 Conclusions

We have evaluated the accuracy and precision of several contemporary reanalysis data sets compared to (a) the COSMIC GPS RO temperatures and (b) the absolute temperature references obtained from the equilibrium properties of certain types

of PSC. In the polar regions, for temperatures below 200 K, the range in the mean biases of the reanalyses with respect to COSMIC temperatures is only $-0.61$ to $+0.48$ K. The lack of assimilated GPS RO data in MERRA is more evident in the higher biases for this data set with respect to COSMIC temperatures in the Antarctic (i.e. where there is a relative paucity of conventional measurements) compared to the other reanalyses. Significantly larger negative biases, except at 100 hPa, and a vertical oscillation are seen in the MLS temperatures. Standard deviations are similar for the reanalyses, but the standard

deviations for MLS temperature are substantially larger.

The extent to which the equilibrium thermodynamics of STS and ice PSCs can be used as absolute temperature references has been explored in detail. The estimated measurement precisions for the STS equilibrium and ice frost points are 0.4 K and 0.3 K, respectively, in the 68–21 hPa pressure range. The corresponding estimated measurement accuracies are in the range 0.7–1.6 K for STS and 0.4–0.7 K for ice. The resulting RMS uncertainites are smaller than those derived for the MLS retrieved

temperatures and comparable to the measurement capabilities of the GPS RO technique (accuracy$<$0.2 K, precision $>$0.7 K) in the lower stratosphere. The reanalysis temperatures were found to be lower than the absolute reference points by 0.3 to 1 K for LIQ and 0 to 1 K for ICE at the peak heights of PSC occurrence (68–32 hPa). Vertical profiles for LIQ show larger negative deviations above 32 hPa than below that level, and also compared to ICE. Medians for LIQ are consistently biased lower than



those for `ICE` by ∼0.5 K. On the 46 hPa pressure level, medians of the reanalyses all depart from zero, and their scatter falls within the range of about 0.6 K for `LIQ` and 0.5 K for `ICE`. Although the biases are larger for `LIQ` than for `ICE`, the standard deviations for `LIQ` (∼0.6 K) are smaller than those for `ICE` (∼0.7 K).

To put these `LIQ` and `ICE` reference temperatures into context with other independent polar temperature measurements, it
is instructive to compare them to the temperature measurement errors from long-duration balloon flights, which have typical nighttime biases of 0.5 K, precisions of 0.4 K (Pommereau et al., 2002) and measured standard deviations of 1.0 to 1.3 K for temperature differences with respect to ECMWF operational analyses (Knudsen et al., 2002).

The polar temperatures from several contemporary reanalyses are in much better agreement than were the reanalyses from previous decades. Hence, in explaining any systematic deficiencies in modeled chlorine activation and/or modeled ozone
losses based on these reanalyses, the burden shifts to finding alternative explanations other than the arbitrary adjustment of the reanalysis temperatures by as much as 1–2 K to offset such discrepancies. However, even though the reanalyis temperature differences are in general in very good agreement, under certain conditions such as wave-driven events, individual temperature comparisons may vary by several kelvin and are therefore important for specific case studies.

## 6    Data availability

MLS data are archived at the NASA Goddard Earth Sciences Data Information and Services Center (Schwartz et al., 2015; Lambert et al., 2015; Manney et al., 2015).

CALIOP data were obtained from the NASA Langley Research Center Atmospheric Science Data Center (CALIOP L1; CALIOP L2).

COSMIC data were obtained from the University Corporation for Atmospheric Research COSMIC Data Analysis and Archive Center (COSMIC)

GEOS5.9.1 data were obtained from the Goddard Earth Sciences Data and Information Services Center (GEOS5.9.1).

MERRA data were obtained from the Goddard Earth Sciences Data and Information Services Center (MERRA).

MERRA-2 data were obtained from the Goddard Earth Sciences Data and Information Services Center (MERRA2).

JRA-55 data were obtained from the Research Data Archive at the National Center for Atmospheric Research, Computational and Information Systems Laboratory JRA-55.





ERA-Interim data were obtained from the Research Data Archive at the National Center for Atmospheric Research, Computational and Information Systems Laboratory (ECMWF)

NCEP-CFSR data were obtained from the Research Data Archive at the National Center for Atmospheric Research, Com-
5    putational and Information Systems Laboratory (NCEP-CFSR).

IDL software for calculation of PSC thermodynamic properties provided by M. E. Hervig was obtained from the GATS
Scientific Software 95 website (http://gwest.gats-inc.com/software/software_page.html).

*Acknowledgements.*    We gratefully acknowledge members of the teams associated with the CALIOP and MLS instruments, and the reanalysis
10    data centers. MLS data are archived at the NASA Goddard Earth Sciences Data Information and Services Center. CALIOP data were obtained
from the NASA Langley Research Center Atmospheric Science Data Center. IDL software for calculation of PSC thermodynamic properties
provided by M. E. Hervig was obtained from the GATS Scientific Software website (http://gwest.gats-inc.com/software/software_page.html).
Work at the Jet Propulsion Laboratory, California Institute of Technology, was carried out under a contract with the National Aeronautics
and Space Administration. This research was conducted as a component of the SPARC (Stratosphere-troposphere Processes And their Role
15    in Climate) S-RIP activity, under the guidance and sponsorship of the World Climate Research Programme.




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





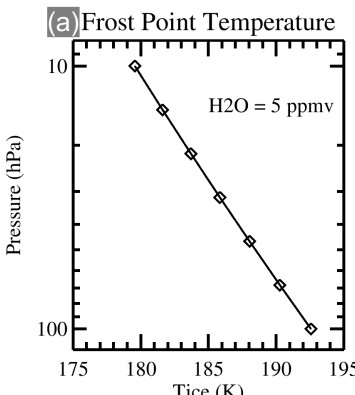
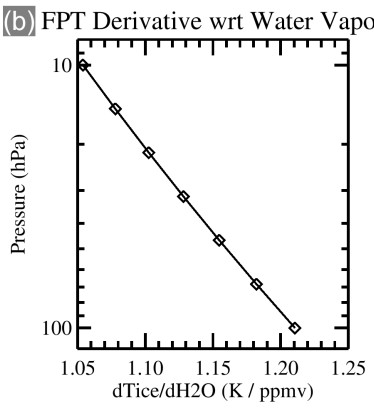
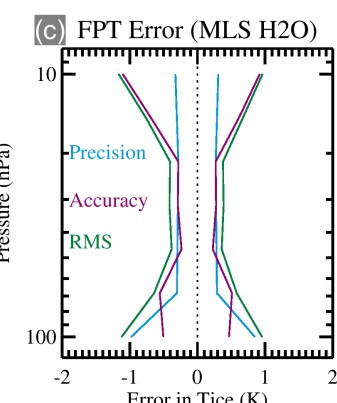

**Figure 1.** (a) Variation of frost point temperature with pressure in the lower stratosphere for a fixed $H_2O$ volume mixing ratio (5 ppmv). (b) Derivative of the frost point temperature with respect to water vapor. (c) Error in determination of frost point temperature ($T_{ice}$) arising from the uncertainties in the MLS $H_2O$ measurement.





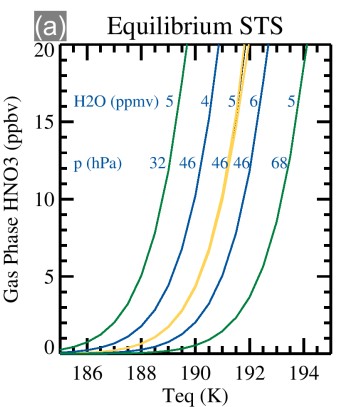
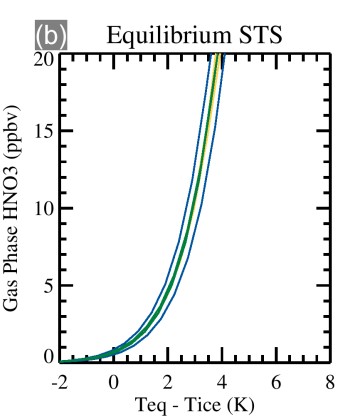
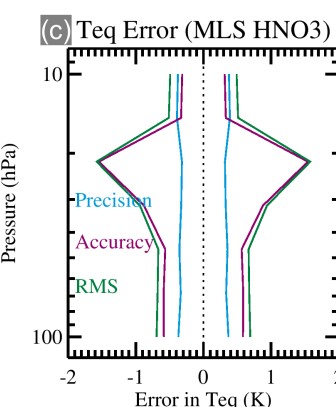

**Figure 2.** (a) Equilibrium $HNO_3$ uptake curves vs temperature for a variety of $H_2O$ mixing ratios (4, 5, 6 ppmv), $H_2SO_4$ mixing ratios (0.1, 0.5 ppbv), and pressures (68, 46, 32 hPa). (b) As for (a), except for a coordinate transformation of the temperature scale relative to the ice frost point ($T_{ice}$). (c) Error in the equilibrium STS temperature arising from the errors in the MLS $HNO_3$.





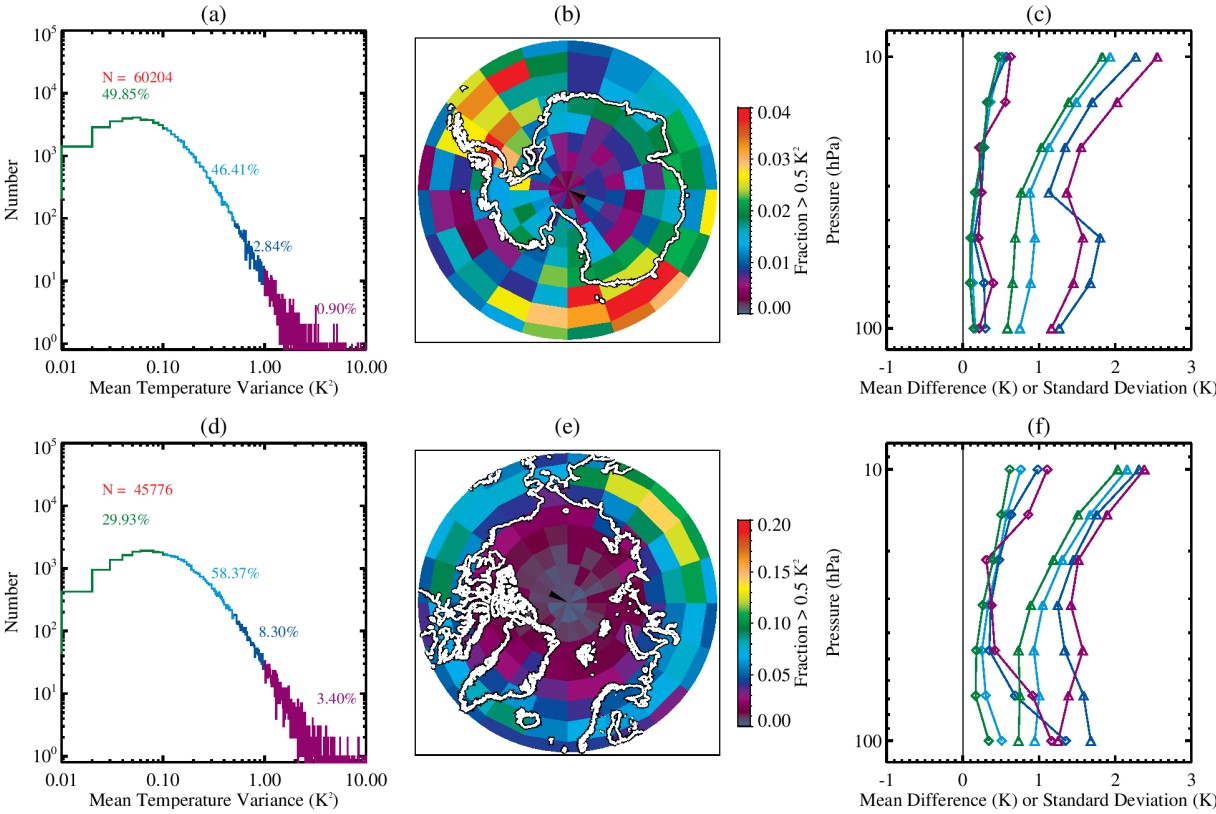

**Figure 3.** Temperature variance, $\sigma^2 = \overline{T'^2}$, in $K^2$, of the COSMIC profiles averaged over the pressure range 80–20 hPa for 2008–2013 for the Antarctic and Arctic. (a,d) Histogram of the mean COSMIC temperature variance with color coding for $\sigma^2 \leq 0.1$ (green), $0.1 > \sigma^2 \leq 0.5$ (cyan), $0.5 > \sigma^2 \leq 1.0$ (blue) and $\sigma^2 \geq 1.0$ (purple). N is the total number of profiles, and the percentages falling in each of the four colored $\sigma^2$ regions are also given. (b,e) Geographic distribution of the fraction of COSMIC profiles with $\sigma^2 > 0.5$. (c,f) Mean values (diamonds) and standard deviations (triangles) of the temperature differences between ERA-I and COSMIC, color coded for the COSMIC temperature variance as given in (a,d).





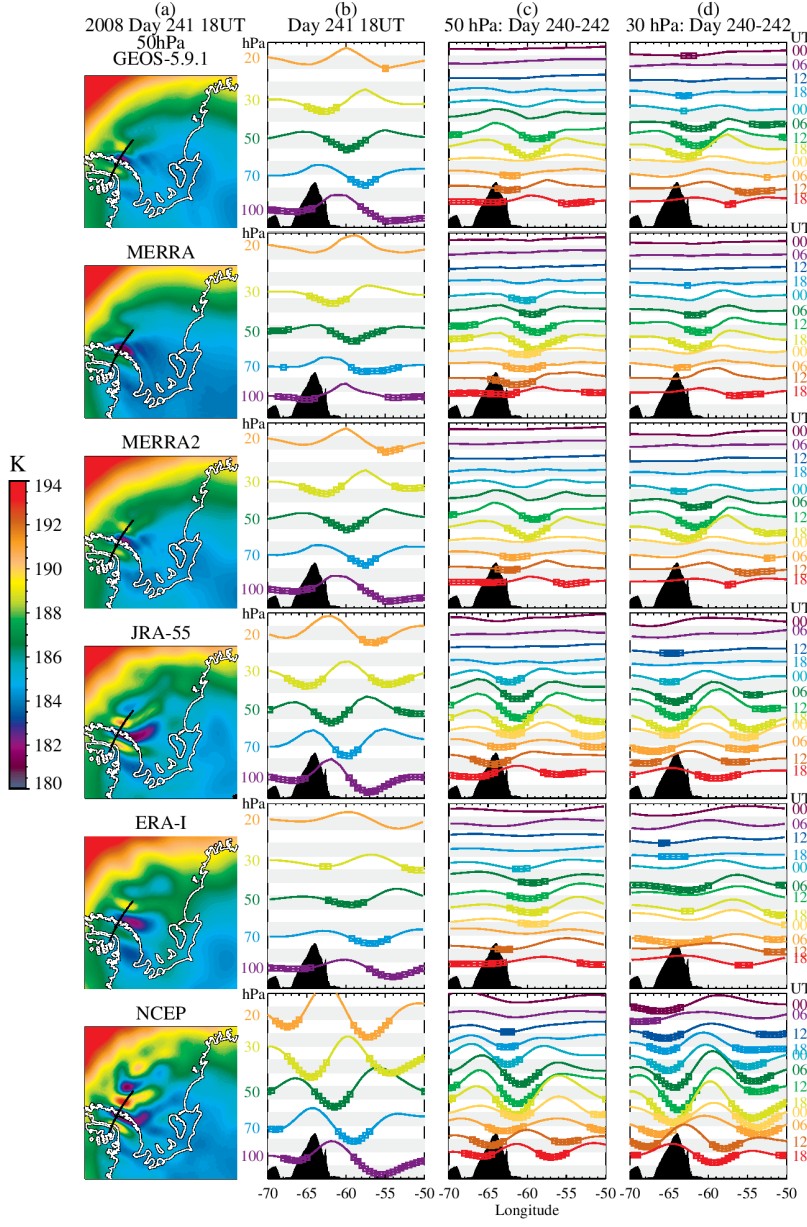

**Figure 4.** (a) Geographic distribution of 50 hPa temperatures over a region of the Palmer Peninsula on 28 August 2008 (day 241) at 18UT for different reanalyses. (b) Longitudinal reanalysis temperature structure from −70° to −50° on the 100, 70, 50, 30, and 20 hPa pressure levels along the 72° S latitude circle (shown as a black line in (a)). The ordinate is an arbitrary temperature scale with offset colored lines representing the longitudinal temperature variation for each of the labeled pressure levels. (c,d) Time series of the 50 hPa (30 hPa) reanalysis temperatures starting from 27 August (d240) at 0UT and ending on 29 August (d242) at 18UT. The ordinate is an arbitrary temperature scale with offset colored lines representing the temperature variation at the 6-hourly intervals labeled on the right. In (b,c,d), temperatures below the ice frost point are highlighted by square symbols, and mountain terrain is indicated by the black shading. Gray/white shading denotes both 5 K temperature bands and 500 m terrain increments.





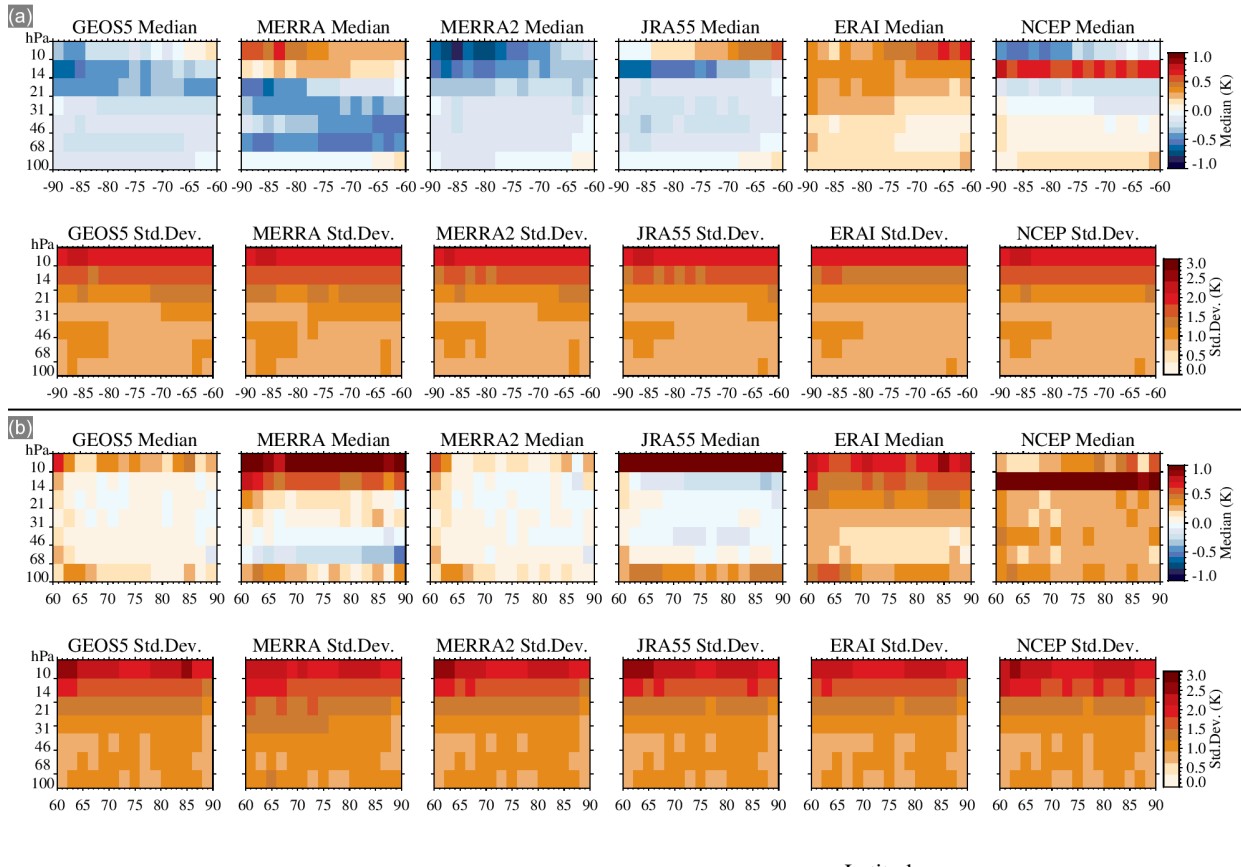

**Figure 5.** Zonal distributions of the median and standard deviations of the differences between the reanalysis temperatures and COSMIC temperatures ($T_{re} - T_{COSMIC\ RO}$) averaged over 2008–2013 for (a) the Antarctic (90° S–60° S) and (b) the Arctic (60° N–90° N).



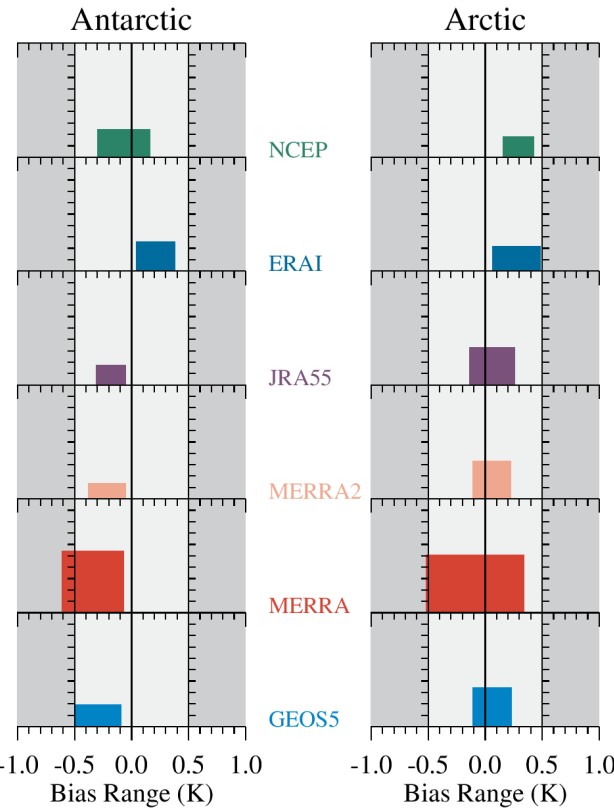

**Figure 6.** Temperature bias ranges of the reanalyses relative to COSMIC GPS RO for Antarctic and Arctic winters 2008–2013, poleward of 60°, and for pressure levels 68–21 hPa, derived from the data in Figure 5. The bias ranges are obtained from the extrema of the individual yearly mean bias values over 68–21 hPa. Each colored box indicates the bias ranges for the particular reanalysis scheme given in the legend; the abscissa gives the range of minimum to maximum bias; the ordinate gives the largest meridional bias range. Light grey shading indicates a ±0.5 K scale in the bias range and a 0–1 K scale for the meridional bias range.





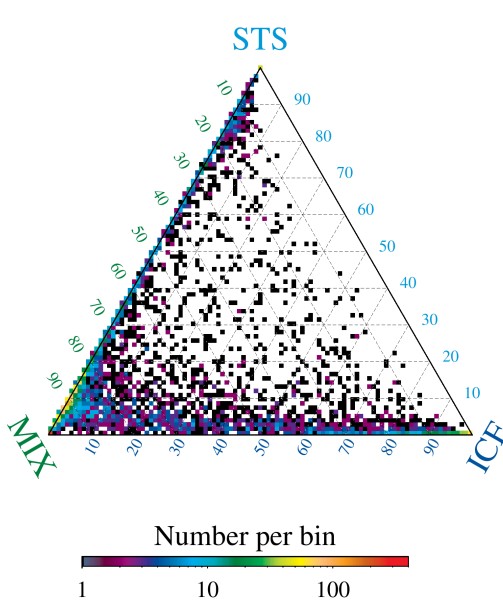

**Figure 7.** Ternary density plot of the relative proportions of `STS/MIX/ICE` occurring in the MLS geometric FOV at temperatures below the ice frost point for the 2013 Antarctic winter at 46 hPa.





**Figure 8.** Along-track vertical cross-sections of PSCs, temperature, HNO$_3$ and H$_2$O over a 7000 km transect crossing Antarctica on 23 June 2008 (2008d175). Column (a) CALIOP PSC types. Column (b) MLS temperature relative to the ice frost point. Column (c) MLS HNO$_3$. Column (d) MLS H$_2$O. Selected regions (black rectangular boxes), centered at 32 hPa in the top row, are shown enlarged in the middle row (1165 km along-track by 4.32 km high), and similarly enlarged again in the bottom row (165 km by 2.16 km). In the bottom row, the resolution is that of the MLS geometric FOV, in which MLS reports only a single measurement value, whereas CALIOP reports ∼400 individual 5 km by 0.180 km pixels.





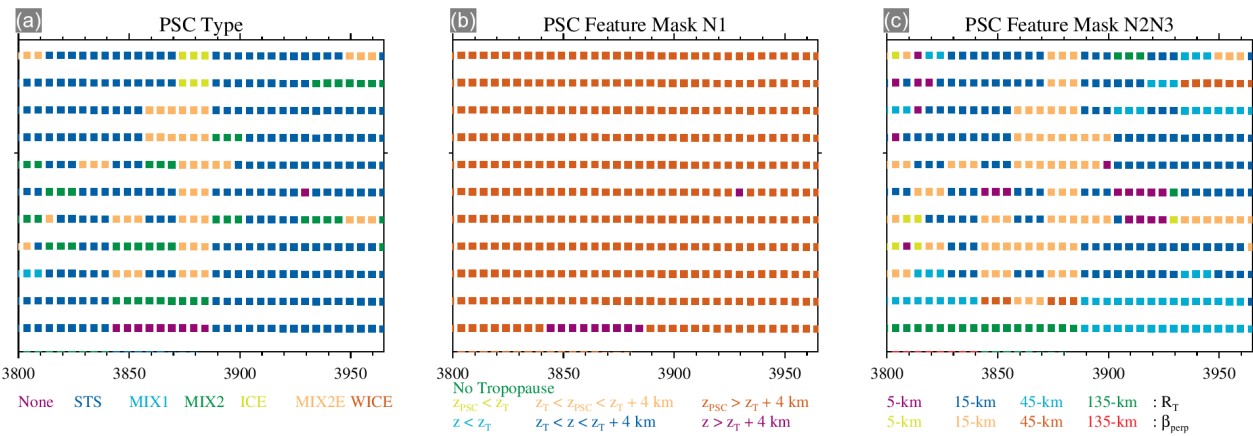

**Figure 9.** Antarctic CALIOP orbit transect corresponding to the 165 km by 2.16 km MLS geometric FOV scenes (bottom row) centered at 32 hPa in Figure 8. (a) CALIOP PSC Type. All types except wave ice are present in this scene consisting of STS:68.9%, ICE:1.6%, MIX1:0.6%, MIX2:12.9%, and MIX2E:13.2%. Clear sky cases (None, purple squares) make up the remaining 2.8%. (b) PSC Feature Mask `N1` encodes the PSC presence and height for each pixel relative to the local tropopause. All PSC detections (orange squares) are at least 4 km above the tropopause. The few pixels containing no PSC detections (purple squares) are also 4 km or more above the local tropopause. (c) PSC Feature Mask `N2N3` encodes the horizontal averaging and backscatter threshold (total backscatter ($R_T$)) or perpendicular ($\beta_{perp}$) used for detection with the color codings given in the legend.





**Figure 10.** Orbit transect from Figure 8 indicating the fraction of the MLS geometric FOV that is filled by each CALIOP classified PSC type and all PSCs. The white areas in the `LIQ` and `ICE` Filled FOV plots indicate the locations that pass the strict FOV filling criteria given in the text.





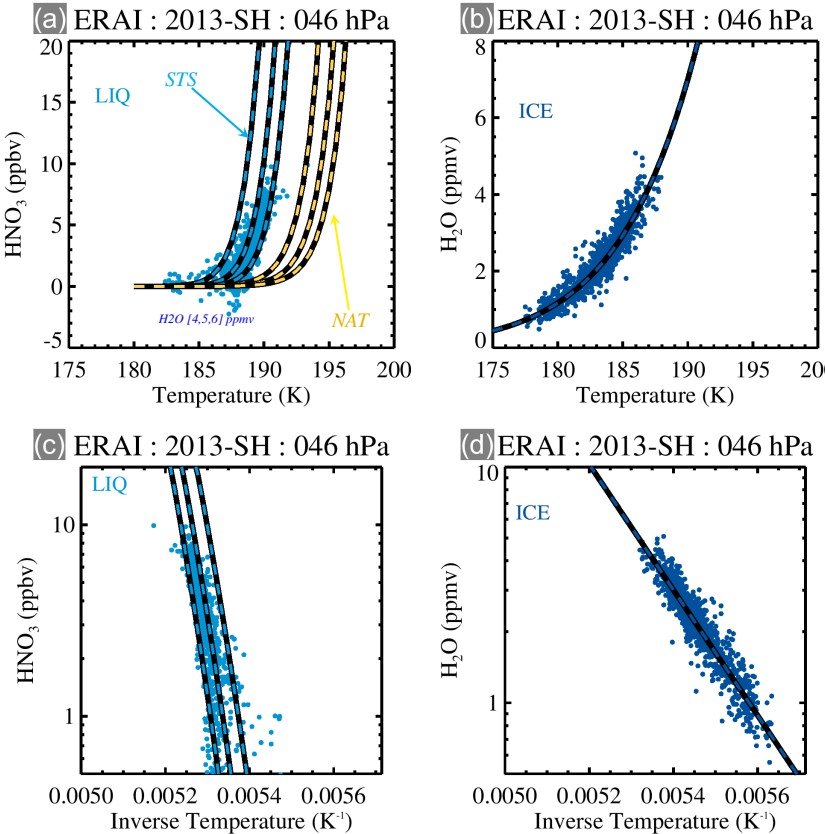

**Figure 11.** Scatter plots of coincident MLS $HNO_3$ and $H_2O$ vs ERA-I temperature for PSCs classified by CALIOP on the 46 hPa pressure level in the 2013 Antarctic winter. (a) MLS $HNO_3$ vs ERA-I temperature for liquid (LIQ) PSCs. The theoretical equilibrium uptake of $HNO_3$ by STS is shown for representative ambient $H_2O$ values by the cyan-black dashed lines. The yellow-black dashed lines show the corresponding NAT equilibrium curves. (b) MLS H2O vs ERA-I temperature for ice (ICE) PSCs. The blue-black dashed lines indicate the theoretical equilibrium for the frost point temperatures. (c,d) As for (a,b), except plotted as log mixing ratio vs inverse temperature.





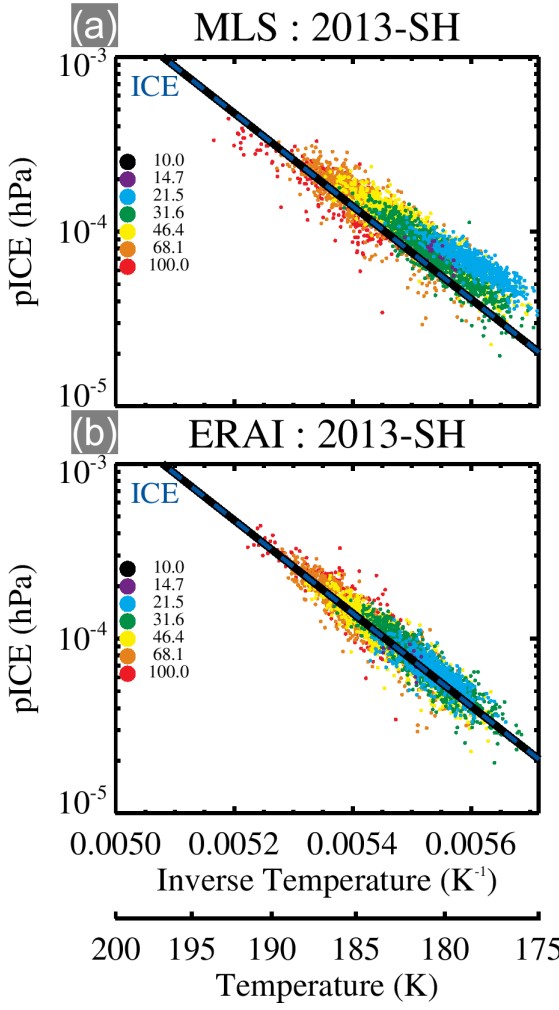

**Figure 12.** Temperature variation of the estimated partial pressure of water vapor over ice, $p_{ice}$, determined from the product of the MLS $H_2O$ volume mixing ratio and the total atmospheric pressure, for Antarctic ice PSCs in 2013. The pressure levels (hPa) are color coded and given in the legend. (a) MLS temperature. (b) ERA-I temperature. The blue-black dashed lines indicate the theoretical equilibrium $p_{ice}$ as a function of the frost point temperature.





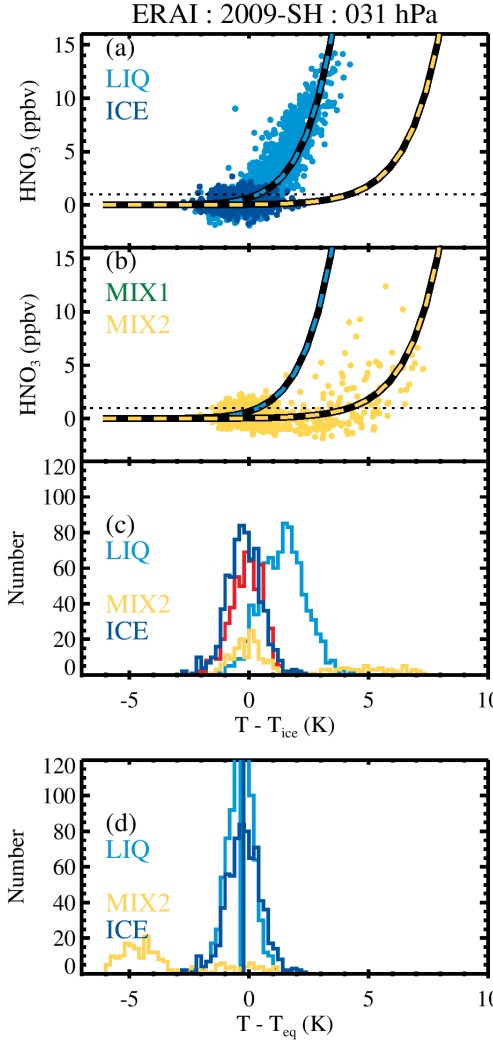

**Figure 13.** Composite statistics for d140–d230 (the bulk of the PSC formation period) of a representative year (2009) of the MLS gas-phase HNO$_3$ variation with ERA-I temperature corresponding to CALIOP PSC classifications at 32 hPa, with the added constraint that at least 75% of the MLS FOV is filled with the same classification. (a) Scatter plot of HNO$_3$ vs temperature deviation from the frost point (T − T$_{ice}$) for PSCs classified as LIQ (cyan) and ICE (blue). (b) As for (a), but for MIX2 PSCs (yellow; because of non-equilibrium effects, which cause larger temperature scatter, MIX2 PSCs are not used in this analysis, but this panel is shown to indicate the good discrimination between the solid and liquid uptake curve branches.) Equilibrium STS (black/cyan dashed) and NAT (black/yellow dashed) curves show the theoretical uptake of HNO$_3$. (c) Temperature histograms for HNO$_3$ mixing ratio >1 ppbv for LIQ PSC type; data in the ICE classification are not subject to this constraint. The red histogram indicates the distribution of LIQ PSCs that have HNO$_3$ below the 1 ppbv threshold. (d) Temperatures transformed according to the STS equilibrium curve for the LIQ classification and NAT equilibrium curve for the MIX2 classification; the ICE classification remains the same as in (c) for comparison.



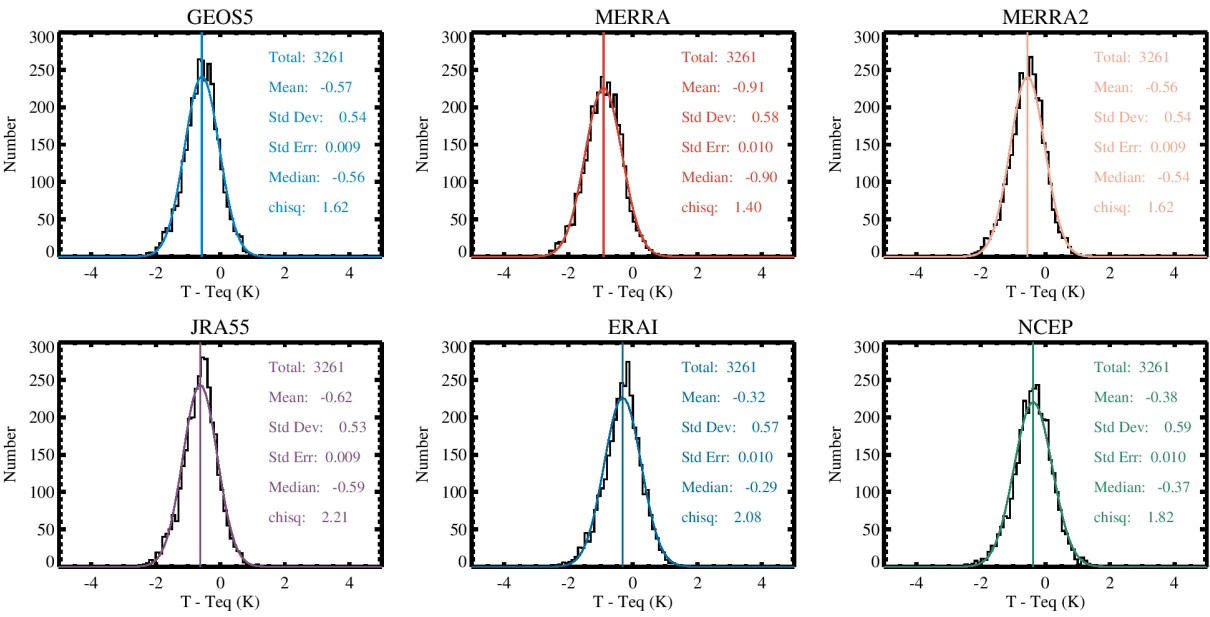

**Figure 14.** Histograms and statistics of the reanalysis temperature distributions for LIQ PSCs relative to the theoretical STS equilibrium temperature ($T_{eq}$) at 46 hPa for the combined 2008-2013 Antarctic winters. A normal distribution curve is superposed on each reanalysis histogram, with the corresponding mean and standard deviation given in the legend.





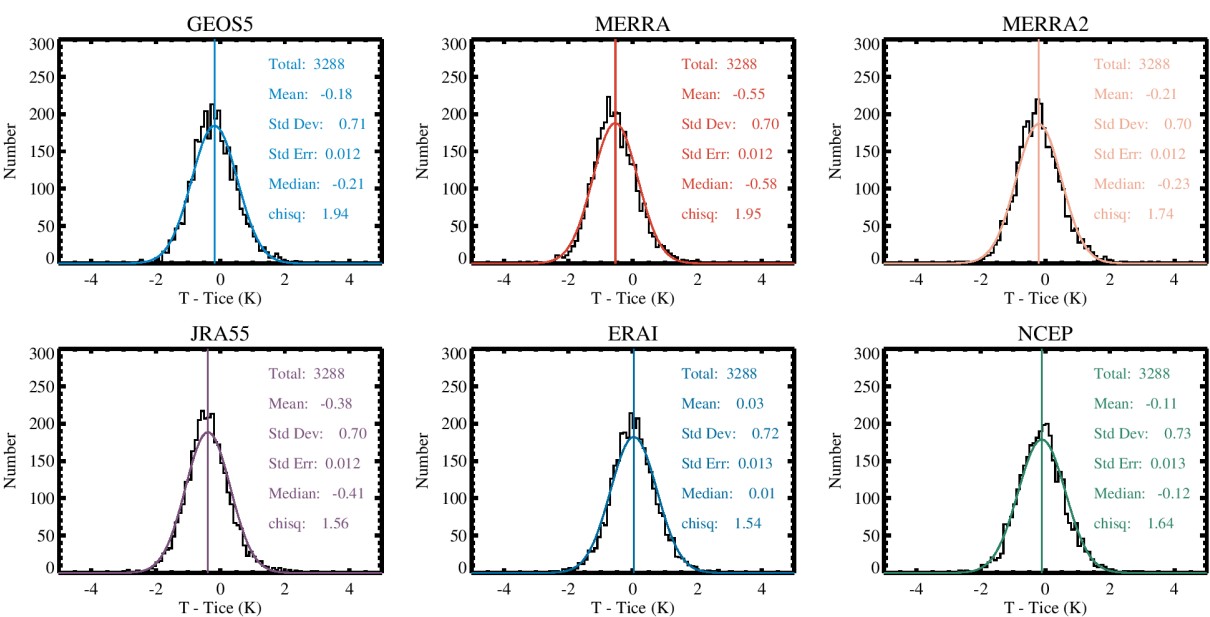

**Figure 15.** As for Figure 14, except for ICE PSCs relative to the theoretical ice frost point temperature.





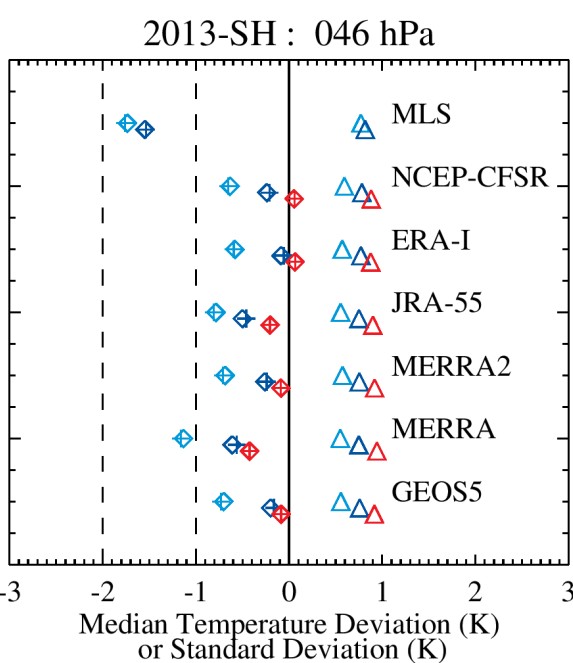

**Figure 16.** Median (diamonds) and standard (triangles) deviations of the temperature distributions for `LIQ` (cyan), `ICE` (blue) and COSMIC (red) for each reanalysis, GEOS-5.9.1, and MLS, for a representative year (2013) at 46 hPa. The plus symbols denote the means of the distributions. Their similarity to the median values indicates that the distributions have small skewness.



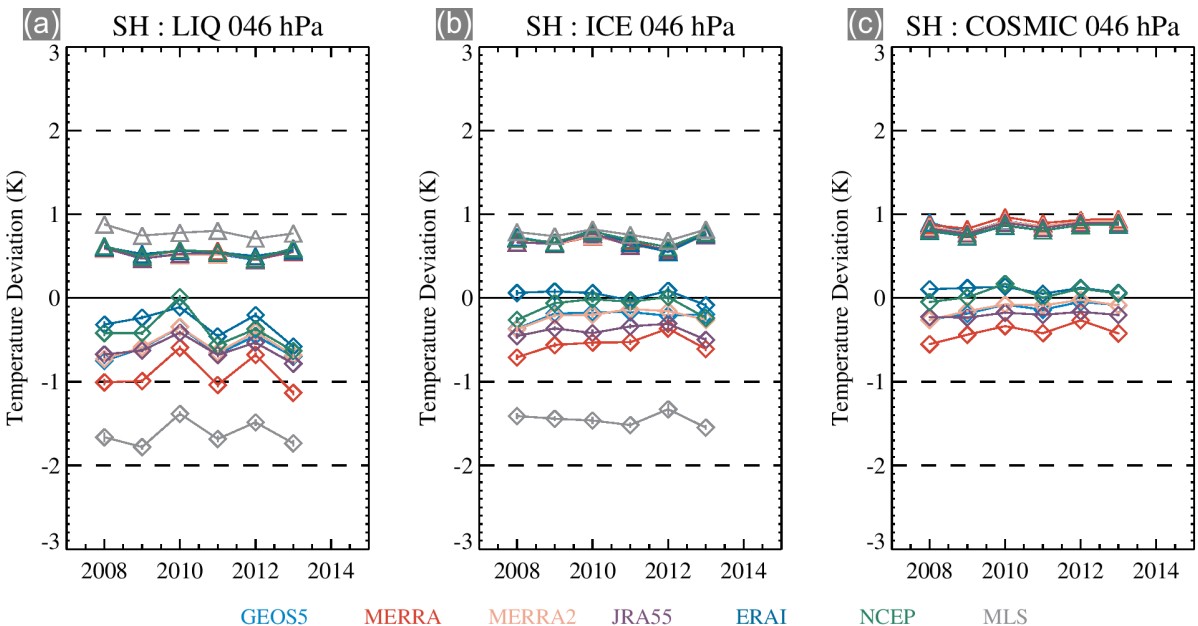

**Figure 17.** Median temperature deviations (diamonds) from $T_{eq}$ for (a) LIQ PSCs and from $T_{ice}$ for (b) ICE PSCs for the temperature distributions in each year 2008–2013 at 46 hPa. The standard deviations of the temperature deviations are also shown (triangles). Lines for the different reanalyses, GEOS-5.9.1, and MLS are color coded (see legend). (c) as for (a,b) except for deviations with respect to COSMIC GPS RO.





**Figure 18.** Vertical profiles of median temperature deviations from $T_{eq}$ for (a) LIQ PSCs and for (b) ICE PSCs for the temperature distributions accumulated over the Antarctic PSC seasons (20 May (d140) to 18 August (d230) from 2008 to 2013). Lines for the different reanalyses, GEOS-5.9.1, and MLS are color coded (see legend); the numerical values on the right-hand side of each panel indicate the total number of observations in the distribution at the corresponding pressure level. (c) as for (a,b) except for deviations with respect to COSMIC GPS RO. (d,e,f) The standard deviations of the corresponding temperature distributions shown in (a,b,c). Dotted lines indicate a standard deviation of 0.5 K



**Figure 19.** As for Figure 18, except for the Arctic PSC seasons (2 December (d336) to 31 March (d090) from 2008/2009 to 2012/2013).





**Figure 20.** Vertical profiles of the temperature differences between (a) LIQ equilibrium and COSMIC GPS RO, $\overline{\Delta T_{LIQ}} - \overline{\Delta T_{COSMIC}}$, and (b) ICE equilibrium and COSMIC RO, $\overline{\Delta T_{ICE}} - \overline{\Delta T_{COSMIC}}$, for the Antarctic. (c,d) As for (a,b) except for the Arctic. Lines for the different reanalyses are color coded according to the legend.





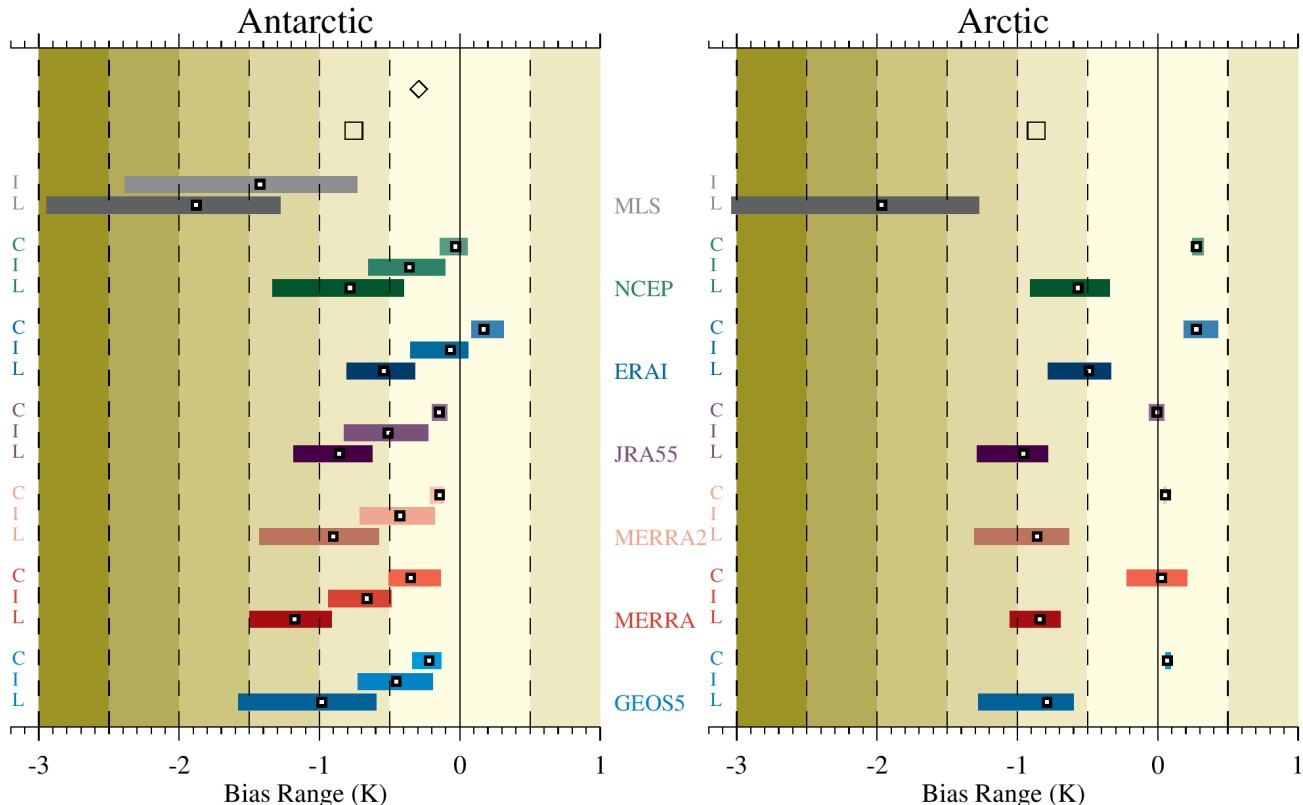

**Figure 21.** Temperature bias ranges of the reanalyses, and MLS, relative to the `LIQ` (L) and `ICE` (I) equilibrium references, and COSMIC (C), for Antarctic and Arctic winters 2008–2013, poleward of 60°, and for pressure levels from 68–21 hPa, derived from the data in Figures 18 and 19. The bias ranges are obtained from the extrema of the weighted yearly mean bias values over 68–21 hPa. Each horizontal colored bar indicates the range of the minimum to maximum bias for MLS and the particular reanalysis scheme given in the legend. White squares with black border indicate the mean bias over 68–21 hPa. Open square (diamond) symbols indicate the mean values of $\overline{\Delta T_{LIQ}} - \overline{\Delta T_{COSMIC}}$ $(\overline{\Delta T_{ICE}} - \overline{\Delta T_{COSMIC}})$ over 68–21 hPa, obtained from Figure 20. There are insufficient statistics for a reliable comparison with the `ICE` reference in the Arctic. Note that MLS has not been compared directly to COSMIC. Background shading indicates 0.5 K increments in the bias range.