# Peer review of "Accuracy and precision of polar lower stratospheric temperatures from reanalyses evaluated from A-train CALIOP and MLS, COSMIC GPS RO, and the equilibrium thermodynamics of supercooled ternary solutions and ice clouds"

_Atmospheric Chemistry and Physics, 2017_

## Referee Comment (RC1) · Anonymous Referee #1 · 11 Aug 2017

General Comments

In this new paper Lambert and Santee evaluate lower stratospheric polar reanalysis temperatures by comparison with CALIOP, MLS, and COSMIC GPS measurements. In particular, the study focuses on the analysis of ice and STS equilibrium temperatures of PSC particles. Overall, I found this to be an interesting and carefully conducted study.

[Figure]

The paper is well-written, concise, and fits in the scope of ACP. I would recommend it for publication once the specific comments listed below have been addressed.

Specific Comments

abstract: The abstract is a bit long and could possibly be shortened. At p2, l21-22 a copyright statement and acknowledgement was introduced, which seems to be out of place?

p3, l13-15: At this point the reader might wonder why NAT is not considered in this study? The reason is given later in the manuscript (at the begin of Sect. 3), but I would like to suggest to move the explanation a bit forward.

p4, l6-10: Regarding the long-duration balloon observations there is a number of other studies using them for evaluation of meteorological reanalyses in the polar stratosphere (some of the papers being part of the recent S-RIP special issue in ACP):

Boccara et al., Accuracy of NCEP/NCAR reanalyses and ECMWF analyses in the lower stratosphere over Antarctica in 2005, J. Geophys. Res.-Atmos., 113, D20115, doi:10.1029/2008jd010116, 2008.

Friedrich et al., A comparison of Loon balloon observations and stratospheric reanalysis products, Atmos. Chem. Phys., 17, 855-866, https://doi.org/10.5194/acp-17-855-2017, 2017

Hoffmann et al., Intercomparison of meteorological analyses and trajectories in the Antarctic lower stratosphere with Concordiasi superpressure balloon observations, Atmos. Chem. Phys., 17, 8045-8061, https://doi.org/10.5194/acp-17-8045-2017, 2017.

I would also suggest to rephrase "Independent datasets such as radiosondes, satellite observations,...", simply because the data sets may have been subject to data assimilation at some of the centers and therefore might not be "independent" in a strict sense.

p5, l9: Add a reference for the MLS measurements?

p5, l10: So far I had not associated ESA with the "A-train" constellation?

p5, l19-30: Maybe summarize this information in a table? It might be worthwhile adding information on the vertical resolution of the data sets in the height range relevant for this study.

p6, l4-5: How was the interpolation performed, linearly with respect to log(p)?

p6, l27-p7, l2: Maybe add a few words regarding the uncertainties of T_STS (or say that this will be discussed later in the paper)?

p7, l11: "Even though the RO measurements _per se_ are SI-traceable..." What does this mean?

p7, l21-31: Not sure if this detailed explanation on how the noise was estimated is really needed? Maybe just summarize the findings of Staten and Reichler?

p8, l1: "...and -0.5K above 35 km" does not seem to be relevant for this study?

p8, l9-12: What are typical values of z_Ttop, z_Sbot, and H_Sbot? Naively, I was thinking z_Ttop and z_Sbot are the same?

p8, l22-28: What is causing the differences between the "dry" and "wet" GPS profiles? The differences do not seem to be related to the water vapor correction at these height levels?

p9, l20: Another prominent reference in addition to Doernbrack et al. is:

Carslaw et al., Increased stratospheric ozone depletion due to mountain-induced atmospheric waves, Nature, 391, 675–678, 1998.

p11, l15-20: The patterns might also be compared with Hindley et al. (2015), who used COSMIC data:

Hindley et al., The southern stratospheric gravity wave hot spot: individual waves and their momentum fluxes measured by COSMIC GPS-RO, Atmos. Chem. Phys., 15,

7797-7818, https://doi.org/10.5194/acp-15-7797-2015, 2015.

p11, 21-32: I found it a bit difficult to follow the text in this paragraph, in particular this sentence: "A decrease in the spread of the temperature differences of reanalysis data compared to COSMIC should be detectable by rejecting the temperature profiles from consideration that are expected to have lower fidelity to the true atmosphere." Maybe rephrase this sentence/paragraph a bit, using easier words?

The message is (as I understood) that you are excluding those COSMIC profiles from the statistical analysis that contain gravity waves, because those are generally not captured well by the reanalyses because of their limited spatial resolution? This makes sense, I think, as it puts emphasis on the evaluation of the large-scale state.

p12, l4: I would suggest to add 1-2 sentences regarding the motivation for the orographic case study at the begin of this section. Reading the paper the first time, I found it a bit strange that at the end of Sect. 4.1 your are saying you are excluding gravity wave-affected data from the analysis and then the next section is showing a gravity wave case study.

Overall, the case study is very interesting, I think, as it clearly illustrates the difficulties related to the representation of gravity waves in reanalyses data. Perhaps the findings could be put into context with other recent work discussing this aspect? The studies listed below might be worthwhile considering. They found that meteorological analyses and forecasts generally tend to show both orographic and non-orographic gravity waves patterns in the correct locations. However, wave amplitudes are significantly underestimated and wavelengths are overestimated (as can be seen also in your Fig. 4), which is attributed to the limited spatial resolution or truncation of the forecast models.

Schroeder et al., Gravity waves resolved in ECMWF and measured by SABER, Geophys. Res. Lett., 36, L10805, doi:10.1029/2008GL037054.

Jewtoukoff, et al., Comparison of Gravity Waves in the Southern Hemisphere Derived

from Balloon Observations and the ECMWF Analyses. J. Atmos. Sci., 72, 3449–3468, https://doi.org/10.1175/JAS-D-14-0324.1, 2015.

Hoffmann et al., A decadal satellite record of gravity wave activity in the lower stratosphere to study polar stratospheric cloud formation, Atmos. Chem. Phys., 17, 2901-2920, https://doi.org/10.5194/acp-17-2901-2017, 2017.

Hoffmann et al., Intercomparison of meteorological analyses and trajectories in the Antarctic lower stratosphere with Concordiasi superpressure balloon observations, Atmos. Chem. Phys., 17, 8045-8061, https://doi.org/10.5194/acp-17-8045-2017, 2017.

p12, l11-15: Which height level are the wind speeds of 25-35 m/s referring to? For estimation of gravity wave vertical wavelengths by means of the dispersion relation the winds at the observational level should be considered. There might be significant Doppler shifting and change in wavelength towards longer wavelengths if the stratospheric winds are stronger than the low-level winds.

p12, l18-20: Are these differences in wave amplitude correlated with the horizontal resolution of the reanalyses data sets?

p13, l11-15: How do these findings regarding the biases of the reanalyses relate to other studies on the topic? Earlier studies found quite significant warm or cold biases of reanalyses data (varying with height) in the polar winter stratosphere, I think.

p13, l18-19: I am not so familiar with the CALIOP classification of PSCs. Do MIX1, MIX2, and MIX2-enh include or are mainly dominated by NAT?

p14, l19-p15, l2: Section 4.6 does not seem to present any results, but recapitulates the method? Maybe merge with Sect. 4.7?

p16, l13-16: I would not directly expect the distance of reanalysis grid points to observations being a reason for causing two distinct modes of temperature precision because I would expect those distances to vary continuously over the study area?

p17, l15-16: I am afraid, do not fully understand this sentence: "The standard deviations result from the combination of the quadrature addition of the precisions of the reanalyses and the reference points." What does "precision of the reference points" refer to?

p18, l24-25: However, the standard deviations with respect to the reanalyses are only similar, because gravity wave-affected profiles showing small-scale temperature fluctuations have been excluded in your analysis, right? It might be good to repeat this here.

p19, l8-9: I would agree that this study showed much better agreement between the reanalyses and observations than earlier work. However, I am not sure if you really demonstrated that point in the paper (see also comment regarding p13, l11-15)? The discussion of this aspect could be extended a bit.

p31, Fig. 3: Maybe indicate the noise variance of the COSMIC data in Figs 3a and 3d? In the caption the unit "Kˆ2" is missing in a few places, e.g. "0.1" -> "0.1 Kˆ2" and so on.

p33, Fig. 5: Does this analysis consider filtering of small-scale fluctuations due to gravity waves? There seem to be notable differences in the medians between the reanalyses. Is this related to COSMIC data being or not being assimilated in some of the reanalysis? How do these results compare to other studies?

Technical Corrections

p11, l4: Remove units "J Kgˆ-1" from equation (3)? And "Kg" should be "kg" in other places?

p12, l10: peninsula -> Peninsula (maybe use either "Palmer Peninsula" or "Antarctic Peninsula" throughout the manuscript for consistency?)

p12, l11: lambda_v -> lambda_z

p14, l31: ERA-I -> ERA-Interim (and in other places)

p36, Fig. 8 and p37, Fig. 9: y-axis labels are partly missing

p44, Fig. 16: fix "standard (triangles) deviations"

—————————————————————

---

## Referee Comment (RC2) · Anonymous Referee #2 · 11 Sep 2017

**Review of ACP manuscript acp-2017-640**

Title: Accuracy and precision of lower stratospheric polar reanalysis temperatures evaluated from A-train CALIOP and MLS, COSMIC GPS RO, and the equilibrium thermodynamics of supercooled ternary solutions and ice clouds

Authors: Alyn Lambert and Michelle L.Santee

The authors present an evaluation of stratospheric temperatures at high latitudes in reanalyses by using temperatures from different satellite based observations including Microwave Limb Sounder (MLS) and GPS radio occultation (RO) measurements.

They furthermore use observed regimes of different types of Polar Stratospheric Clouds (PCSs) for deriving temperature references based on the thermodynamics of supercooled ternary solutions and ice clouds.

The paper is well written but overall it is quite lengthy and includes 21 figures. At several places the content could easily be condensed and streamlined without losing information and thus presented in a more compact way. A sincere restructuring of the text in some sections is recommended and a restructuring/merging of figures. Furthermore, clarifications are needed regarding the definition of precision and accuracy, the choice of pressure levels, and the description of comparison methods (please see comments below).

I recommend publishing the manuscript after revision taking into account the reviewer's recommendations and providing necessary clarifications. Please find a list of major and minor comments below.

**Major comments:**

**Abstract:** The abstract is much too long and definitely should be shortened by half. Only essential information and results should be stated. E.g., line 10 to 16 on page 2 could be deleted.

Is the statement on GPS RO uncertainties correct: "accuracy <0.2 K, precision > 0.7K" (also in paper text and conclusion section)? To my knowledge it should be vice versa. The precision is about 0.25 K and the accuracy in the tropopause region about 0.7 K increasing exponentially in the stratosphere (see also comment below).

**Methods:**
Computation of pressure levels:
"The synoptic gridded reanalysis temperatures are interpolated to a common vertical pressure grid (100–10 hPa), with 6 levels per decade …"
Why are the pressure levels chosen this way for the reanalyses and the radio occultation observations, e.g., levels of 61 hPa or 46 hPa, etc.? Is there a specific reason for this? I strongly recommend using standard pressure levels or pressure levels with 10 hPa steps, i.e., plain numbers, as used later in the manuscript anyway. What does "fixed pressure levels per decade" mean?
"For convenience we interpolated the COSMIC temperature profiles to a set of fixed pressure levels with 24 levels per decade …., approximately 2/3 km vertical resolution. "
Please revise the last statement.
The vertical resolution of GPS RO observations ranges from a 0.5 km (100 m in case of wave optics processing) in the troposphere to about 1.5 km in the stratosphere. By interpolating the vertical levels you change the sampling of the height/pressure levels.

Section 2.4:
Please explain explicitly how the reanalyses are compared with the radio occultation observations. As I understand you compare the gridded reanalyses data with individual COSMIC temperature profiles (supposedly averaged in the respective grid or zonal mean). Doing it this way you would not account

for the different sampling of the observations. A comparison should be either performed by subsampling the reanalyses at the locations of the RO observations (i.e., comparing collocated profiles) or by comparing monthly-mean zonal-mean reanalyses data with zonal-mean sampling-error corrected RO climatologies

Cite Schreiner et al. (2007; doi:10.1029/2006GL027557) in section 2.4 as they made the first analysis on the precision of COSMIC data. Their results were confirmed by Alexander et al. (2014), who derived a precision of about 0.1% in refractivity between 8-25 km (larger above 30 km) which translates to about 0.25 K. No strong dependence of the estimated precision on latitude or season was found.
Is this consistent with your statement in the abstract and conclusion section on precision and accuracy?

Please include a definition of precision and accuracy in the method section before values for the different data sets are discussed and give an explicit description on how precision and accuracy are computed in your study.
Move method descriptions from the results section to the method section (see comments below on restructuring).

**Structure of the manuscript:**
Introduction:
Page 4, 3$^{rd}$ paragraph: This paragraph already describes the methodological approach in some detail and would rather fit at the beginning of the method section.
Page 4, line 29: "However, lack of sufficient statistics in the Arctic precludes a robust conclusion for ICE PSCs." This is a result and the sentence should be moved to the conclusion section or/and the results section.
Page 4, line 31-33: Detailed description of investigated dates should be moved to the beginning of the methods section.

Method section:
I recommend slight restructuring: Start with the reanalyses description. The information in the first paragraph does not really fit here at the beginning and should be moved to sections 2.2 where CALIOP observations are described and to section 2.3 where MLS is described.

Section 2.1: The information on all the reanalyses could be summarized in a Table and would give a more compact overview.

Streamline page 7 and 8 as you recapitulate several studies in quite a lengthy way.

Results sections:
The results sections include method descriptions such as section 4.6 describing how the comparison of reanalyses temperatures to LIQ and ICE reference points is performed and in section 4.8.2 (last paragraph) on difference computations. These method descriptions should be moved to the method section.

Furthermore some results sections are very short and could be merged into one section such as sections 4.4, 4.5, and 4.7. There is no need to make a separate section for each figure but rather to comprise related topics in one section. The merged section 4.4 could be entitled, e.g., "PSC types from CALIOP and their representation in MLS observations"

Section 4.8 Polar temperature reference points: – remove sub-sub-sections and streamline the text. Section title could be "Reanalysis temperatures compared to LIC and ICE temperature reference points".

**Figures:**
General: You often provide the same information twice in the figure captions and in the description of figures in the text. Please give the technical description of the figure in the caption and provide explanations and description of the scientific results in the text, e.g., Fig. 13.

Fig. 5: The latitudinal structure from 60S/N to 90S/N is rather uniform for the median and the standard deviation. A structure in temperature differences is mainly visible in the vertical domain. Averaging over 60S-90S and 60N-90N would give a condensed overview. Median and standard deviation profile could be shown in one panel. Fig. 5 could be condensed in 2 rows for the Arctic and Antarctic. In Figure 6 the information on latitudinal differences and vertical differences is given again, anyway.

Fig. 6: The vertical boxes should also be centered around a zero line as done for the latitudes.

Fig. 9: Please condense information. Fig.9a is the same Figure as already shown in Fig. 8. Fig.9b does not really give much information, it can be stated in the text. Please check whether this figure is really needed or the information could be stated in the text.

Fig. 11: What is the reason for showing the information twice? Why do you show inverse temperature? It is distracting to the reader unless there is a specific reason for it. I suggest plotting a logarithmic y-axis, using temperature for the x-axis, and showing only 2 plots in Fig. 11.

Fig. 12: Suggest having a similar design as for Fig. 11, i.e., plotting the temperature axis from 175 K to 200 K and removing the inverse temperature x-axis.

Fig. 18 and 19: The standard deviations (Fig. 18.d-f) could be easily plotted in Fig. 18 a-c by extending the x-axis to -3K to 3K. The same could be done for Fig. 19. Fig. 18 and 19 could then be merged to one Fig. showing SH in the upper panels and NH in the lower panels.
So the reader would be provided with compact information.

Fig. 14 and 15:
Please compile the information printed in the different panels in a Table. Fig. 14 and 15 could be removed and information of section 4.8.1 could be streamlined. The description at page 16, line 10 to 18, could be removed and the main information stated in one sentence.

**Minor comments:**

The title is very long and contains 5 acronyms. It should be shortened. I think it is not necessary to list all the sensors explicitly in the title, could be replaced by "satellite observations". Also the term "supercooled ternary solutions" might not be a familiar term for many readers. Maybe use "polar stratospheric cloud types or polar stratospheric cloud regimes" instead.

Reanalysis: More than one reanalysis is evaluated, rather use "reanalyses" in the title.

Page 2, line 19: "significant low bias in MLS temperature of up to 3 K" ? Do you mean "large bias"?

Page 2, line 31: What does "(one-off research satellites)" mean, check singular/plural.

Page 2, line 32: What is "temperatures measured with the Geophysica": Please be explicit. What is the Geophysica, how were the temperatures measured, state the instrument?

Page 6, line 26: "-2-0 K" change to "-2 K to 0 K"

Page 7, line13: "requires a number of assumptions because of the long ray path through a non-uniform atmosphere. Therefore, corrections are required for ionospheric effects, variations in water

vapor, and gradients in in temperature along the ray path."
This statement is not fully correct. Please reformulate/clarify.
Initialization (of bending angles) at high altitudes is performed where the signal to noise ratio of the measurement becomes low. An ionospheric correction is applied through differencing of the two GPS frequencies to remove the ionospheric contribution to the measurement and to gain information on the neutral atmosphere only. Refractivity then includes information on the dry and moist part of the atmosphere (see Smith and Weintraub formula, as you explain on the next page). In a dry atmosphere, temperature can be retrieved without any further correction or initialization. In a moist atmosphere, i.e., in the mid- to lower troposphere, additional background information is needed to separate temperature and humidity information (usually through a 1DVar retrieval).

However, as you investigate temperatures at high latitudes in the stratosphere, you can use dry temperatures only. The discussion on wetPrf temperatures can be removed as they are not used.

Page 7: line 34: Remove "similar results were obtained over data sparse (Pacific) and data dense (USA) regions". This is distracting as one might think it is RO data sparse and dense regions (but RO has almost equal distribution globally) but actually it relates to observations assimilated into ECMWF.

Page 8, line 21,22: insert "about" before "14 km" and before "9 km".

Page 9, line 20: Why a pressure range of 68 to 21 hPa and not 70 to 20 hPa.

Page 10, Eq.3: correct to "J kg$^{-1}$" and through paper text.

Page 11: "A decrease in the spread of the temperature differences of reanalysis data compared to COSMIC should be detectable by rejecting the temperature profiles from consideration that are expected to have lower fidelity to the true atmosphere."
What is meant by fidelity to the true atmosphere? The truth is actually not known.
Rejecting observational profiles with higher variance will of course improve the difference between reanalyses and observations if observations show a higher variance than the reanalyses at high latitudes, which is well known for RO. I actually do not understand the purpose of rejecting profiles with higher variance.

Page 12, line 28: What do you mean with "the geophysical structure"? The vertical structure versus the latitudinal structure at high latitudes is shown.

Page 13, line 2: Use clear wording in description of Fig. 6. Rather than saying the "box heights" it is the differences in the vertical between reanalyses and RO.

Page 13, line 22: Why do you show the 46 hPa level?

Page 14, end of section 4.5: Please give percentage of scenes analyzed with respect to available scenes.

Page 17: The last part of section 4.8.2 (page 18, first paragraph) is very hard to read. Please reformulate and streamline info.

Page 44, Fig. 16, caption: Correct to: "Median (diamonds) and mean (triangles)…"

Section titles: Please remove Tre-Tro, Tre-Teq, and similar in section titles.

---

## Author Comment (AC1) · 22 Nov 2017

**acpd-2017-640 Reply to Referee Comments**

**Accuracy and precision of lower stratospheric polar reanalysis temperatures evaluated from A-train CALIOP and MLS, COSMIC GPS RO, and the equilibrium thermodynamics of supercooled ternary solutions and ice clouds**

**A. Lambert and M. L. Santee**

We thank the reviewers for their careful reading of the manuscript and appreciate their suggestions for improvements. We address their specific comments in the following text and outline the expected changes to the revised manuscript accordingly.

We have removed 3 of the 21 figures in the original manuscript (Figures 9, 14 and 15 in previous manuscript numbering).

The revised manuscript has 2 new figures (in response to referee comments) for a total of 20 figures (Figures 3 and 18 in revised manuscript numbering).

Referee Comments RC1/RC2 are in blue type.

Author Responses are in brown type.

**RC1 General Comments:**

In this new paper Lambert and Santee evaluate lower stratospheric polar reanalysis temperatures by comparison with CALIOP, MLS, and COSMIC GPS measurements. In particular, the study focuses on the analysis of ice and STS equilibrium temperatures of PSC particles. Overall, I found this to be an interesting and carefully conducted study.

The paper is well-written, concise, and fits in the scope of ACP. I would recommend it for publication once the specific comments listed below have been addressed.

We will address the concerns of the referee in the modified manuscript. We reply to specific comments below.

**RC1 Specific Comments:**

abstract: The abstract is a bit long and could possibly be shortened. At p2, l21-22 a copyright statement and acknowledgement was introduced, which seems to be out of place?

We agree that the abstract is a bit long and it has been shortened. Copyright and acknowledgement were placed following a request by the editorial support office.

p3, L13-15: At this point the reader might wonder why NAT is not considered in this study? The reason is given later in the manuscript (at the begin of Sect. 3), but I would like to suggest to move the explanation a bit forward.

A brief explanation was added.

p4, L6-10: Regarding the long-duration balloon observations there is a number of other studies using them for evaluation of meteorological reanalyses in the polar stratosphere (some of the papers being part of the recent S-RIP special issue in ACP):

Boccara et al., Accuracy of NCEP/NCAR reanalyses and ECMWF analyses in the lower stratosphere over Antarctica in 2005, J. Geophys. Res.-Atmos., 113, D20115, doi:10.1029/2008jd010116, 2008.

Friedrich et al., A comparison of Loon balloon observations and stratospheric reanalysis products, Atmos. Chem. Phys., 17, 855-866, https://doi.org/10.5194/acp-17-855- 2017, 2017.

Hoffmann et al., Intercomparison of meteorological analyses and trajectories in the Antarctic lower stratosphere with Concordiasi superpressure balloon observations, Atmos. Chem. Phys., 17, 8045-8061, https://doi.org/10.5194/acp-17-8045-2017, 2017.

Boccara et al. did not investigate the reanalyses we used here.

Friedrich et al. reported that "Loon temperature measurements are not suitable for comparison with reanalyses".

We have added a discussion of the Hoffmann et al results (not published at the time we submitted our manuscript) and Figure 18 to facilitate the discussion at the end of the results section (see below).

I would also suggest to rephrase "Independent datasets such as radiosondes, satellite observations,...", simply because the data sets may have been subject to data assimilation at some of the centers and therefore might not be "independent" in a strict sense.

Data centers have also run "data denial experiments" which are independent in a strict sense so we have added an example for that process.

p5, L9: Add a reference for the MLS measurements?

We added Waters et al (2006) (already cited later in the paper).

p5, L10: So far I had not associated ESA with the "A-train" constellation?

We modified the text to refer simply to the "A-train". "NASA", "CNES" and "JAXA" are the space agencies responsible for coordination of the satellites in the A-Train. e.g. see `https://www.nasa.gov/mission_pages/a-train/a-train.html`.

p5, L19-30: Maybe summarize this information in a table? It might be worthwhile adding information on the vertical resolution of the data sets in the height range relevant for this study.

We have provided a table. We have added the vertical spacings for the reanalyses.

p6, L4-5: How was the interpolation performed, linearly with respect to log(p)?

Yes, this is now indicated in the revised text.

p6, L27-p7, L2: Maybe add a few words regarding the uncertainties of T_STS (or say that this will be discussed later in the paper)?

We added the following text. "Estimated uncertainties in the reference point temperatures are discussed in Sections ..."

p7, L11: "Even though the RO measurements 'per se' are SI-traceable..." What does this mean?

The RO measurements are timing measurements of the phase delays of the signal along the atmospheric path from the GNSS transmitter to the LEO receiver. The phase delays are accurately calculated by reference to the international atomic time standard and as such are traceable to the SI second. We modified the text. "Even though the RO measurements of signal phase delays are traceable to the SI second, ..."

p7, L21-31: Not sure if this detailed explanation on how the noise was estimated is really needed? Maybe just summarize the findings of Staten and Reichler?

We prefer to retain this level of detail.

p8, L1: "...and -0.5K above 35 km" does not seem to be relevant for this study?

Although not directly relevant, it does no harm either to point out that the RO accuracy becomes worse above the region studied.

p8, L9-12: What are typical values of z_Ttop, z_Sbot, and H_Sbot? Naively, I was thinking z_Ttop and z_Sbot are the same?

z_Ttop is the top of the tropopause whereas z_Sbot is the bottom of the stratosphere and the gap in between is the tropopause transition layer (TTL). Typical values vary with latitude/season. Only z_Sbot is relevant for our study. We give values we found in the results section.

p8, L22-28: What is causing the differences between the "dry" and "wet" GPS profiles? The differences do not seem to be related to the water vapor correction at these height levels?

We used the "dry" profiles since they are appropriate for the extremely low moisture content of the atmospheric region above 100 hPa. However, since other studies have used "wet" instead of "dry" profiles in the same region we decided to report on the differences we found. Although we do not know the origin of the differences, the point is made that the differences are small enough that either "wet" or "dry" could have been used for this analysis... "The wet temperatures yield results that are consistent with the dry temperatures and therefore are not considered further in this paper."

p9, L20: Another prominent reference in addition to Doernbrack et al. is: Carslaw et al., Increased stratospheric ozone depletion due to mountain-induced atmospheric waves, Nature, 391, 675678, 1998.

We added the reference to Carslaw et al.

p11, L15-20: The patterns might also be compared with Hindley et al. (2015), who used COSMIC data:

Hindley et al., The southern stratospheric gravity wave hot spot: individual waves and their momentum fluxes measured by COSMIC GPS-RO, Atmos. Chem. Phys., 15, 7797-7818, https://doi.org/10.5194/acp-15-7797-2015, 2015.

We added the reference to Hindley et al.

p11, L21-32: I found it a bit difficult to follow the text in this paragraph, in particular this sentence: "A decrease in the spread of the temperature differences of reanalysis data compared to COSMIC should be detectable by rejecting the temperature profiles from consideration that are expected to have lower fidelity to the true atmosphere." Maybe rephrase this sentence/paragraph a bit, using easier words? The message is (as I understood) that you are excluding those COSMIC profiles from the statistical analysis that contain gravity waves, because those are generally not captured well by the reanalyses because of their limited spatial resolution? This makes sense, I think, as it puts emphasis on the evaluation of the large-scale state.

Yes, you understood our intent correctly. We modified the explanation in the revised text as suggested. We also reference the Hoffmann et al (2017) study of Concordiasi balloon data.

"Small-scale temperature fluctuations cannot be captured accurately by the reanalysis data because of their limited spatial resolution. For example, a study of superpressure balloon measurements by Hoffmann et al (2017a) found that ERA-I, MERRA and MERRA-2 reproduced about 30% of the standard deviation of the balloon temperature fluctuations, whereas the lower spatial resolution NCEP/NCAR reanalysis reproduced only 15% and the higher resolution ECMWF operational analysis reproduced 60%. ECMWF analyses also underestimate both gravity wave momentum fluxes derived from the balloon measurements (Jewtoukoff et al., 2015) and wave amplitudes derived from Aqua AIRS (Hoffmann et al., 2017b). Similarly, the COSMIC data should perform better than the reanalyses because of their higher resolution."

p12, L4: I would suggest to add 1-2 sentences regarding the motivation for the orographic case study at the begin of this section. Reading the paper the first time, I found it a bit strange that at the end of Sect. 4.1 your are saying you are excluding gravity wave-affected data from the analysis and then the next section is showing a gravity wave case study.

A brief statement was added.

Overall, the case study is very interesting, I think, as it clearly illustrates the difficulties related to the representation of gravity waves in reanalyses data. Perhaps the findings could be put into context with other recent work discussing this aspect? The studies listed below might be worthwhile considering. They found that meteorological analyses and forecasts generally tend to show both orographic and non-orographic gravity waves patterns in the correct locations. However, wave amplitudes are significantly underestimated and wavelengths are overestimated (as can be seen also in your Fig. 4), which is attributed to the limited spatial resolution or truncation of the forecast models.

Schroeder et al., Gravity waves resolved in ECMWF and measured by SABER, Geophys. Res. Lett., 36, L10805, doi:10.1029/2008GL037054.

Jewtoukoff, et al., Comparison of Gravity Waves in the Southern Hemisphere Derived from Balloon Observations and the ECMWF Analyses. J. Atmos. Sci., 72, 34493468, https://doi.org/10.1175/JAS-D-14-0324.1, 2015.

Hoffmann et al., A decadal satellite record of gravity wave activity in the lower stratosphere to study polar stratospheric cloud formation, Atmos. Chem. Phys., 17, 2901- 2920, https://doi.org/10.5194/acp-17-2901-2017, 2017.

Hoffmann et al., Intercomparison of meteorological analyses and trajectories in the Antarctic lower stratosphere with Concordiasi superpressure balloon observations, Atmos. Chem. Phys., 17, 8045-8061, https://doi.org/10.5194/acp-17-8045-2017, 2017.

The case study was chosen simply based on a striking observation of correlated ice PSCs and cold phases. The responses of the reanalyses shown in Fig 4 do vary to a surprising extent and, although we should be wary of general conclusions based on one sample, their spatial resolutions appear to be correlated in the way expected.

We have added a brief discussion in the text concerning the points you raised...

"In the rest of this paper we concentrate on the large-scale temperatures, facilitated by the effective low-pass filtering using the $E_p$ threshold to remove gravity-wave fluctuations. However, further work on issues related to the representation of gravity waves in the reanalyses would be worthwhile, and the high-pass side of the $E_p$ threshold may provide useful insights."

p12, L11-15: Which height level are the wind speeds of 25-35 m/s referring to? For estimation of gravity wave vertical wavelengths by means of the dispersion relation the winds at the observational level should be considered. There might be significant Doppler shifting and change in wavelength towards longer wavelengths if the stratospheric winds are stronger than the low-level winds.

We modified the text to indicate the height is that of the ice PSCs. "... the wind direction is almost orthogonal to the ridge line, with wind speeds $u \sim 25$–$35$ ms$^{-1}$ corresponding to the observation of ice clouds in the pressure range 80–40 hPa."

p12, L18-20: Are these differences in wave amplitude correlated with the horizontal resolution of the reanalyses data sets?

Yes, NCEP-CFSR has the highest horizontal resolution and ERA-I the lowest compared to the other reanalyses. We have modified the text accordingly.

"At 50 hPa the peak to trough temperature differences are 7.4, 6.2. 8.0, 10.0, 6.1 and 14.9 K for GEOS-5.9.1, MERRA, MERRA-2, JRA-55, ERA-I and NCEP-CFSR, respectively; i.e., over a factor of two difference is seen in the temperature amplitudes between the reanalyses. These differences in the wave amplitudes are rather well correlated with the spatial resolutions of the reanalyses (see Table 1)."

p13, L11-15: How do these findings regarding the biases of the reanalyses relate to other studies on the topic? Earlier studies found quite significant warm or cold biases of reanalyses data (varying with height) in the polar winter stratosphere, I think.

This is now discussed at the end of the results section.

"Recently, Hoffmann et al. (2017a) used superpressure balloon measurements made during the Antarctic Concordiasi campaign in September 2010 to January 2011 to validate meteorological analyses and reanalyses. Over the flight paths of the balloons (17–18.5 km (58.2–69.1 hPa) and $60°$ S – $85°$ S), Hoffmann et al. found warm biases ranging from 0.6 to 1.5 K for ERA-I, MERRA and MERRA-2, increasing towards the pole, and standard deviations of 0.5–0.8 K, with ERA-I showing smaller standard deviations than MERRA. Whereas Hoffmann et al. find MERRA to be the warmest and ERA-I the coldest, we find the opposite order compared to the

COSMIC and thermodynamic temperature references (see Figure 16). Also we find negative temperature biases for MERRA and MERRA-2 and only a slight positive bias for ERA-I, with no significant latitudinal gradients (Figure 6). To reconcile this apparent discrepancy we show daily mean temperatures (at 12UT) for MERRA and MERRA-2 relative to ERA-I in Figure 18 and highlight the non-overlapping time periods of the PSC analysis window (green line) and the balloon flights (red line). Clearly, measurements in the later time period of the balloon flights (September-December) sample different atmospheric conditions than experienced in the earlier time period (May-August). Differences between reanalysis temperatures along the individual balloon trajectories are likely to be amplified compared to the differences in mean polar cap temperatures.

As we noted previously MERRA does not assimilate COSMIC data, whereas MERRA-2 and the other reanalyses we investigated do assimilate COSMIC data, and hence some of the reduction in the bias of MERRA-2 compared to MERRA seen in Figure 18 is likely to be attributable to the use of GPS RO data in the former.

Large vertical oscillations of up to $\pm2$–3 K were reported in the differences between ECMWF operational analyses and the CHallenging Minisatellite Payload (CHAMP) RO data in the 2002 to 2006 June-August Antarctic polar vortices (Gobiet et al (2005), Gobiet et al (2007)). Similar height-dependent features were also found (Parrondo et al 2007) in comparisons of ECMWF analyses with radiosondes. The value of the RO dataset as a reference was ably demonstrated by comparisons of CHAMP RO with ECMWF and other meteorological data (GEOS-4, MetO, NCEP/CPC, NCEP Rean) for June-August 2003 (Gobiet et al 2007). None of the other dataset differences showed features consistent with ECMWF, nor were any two consistent with each other, e.g. GEOS-4 displayed an Antarctic cold bias relative to CHAMP at 25–30 km, whereas NCEP Rean displayed a warm bias in roughly the same range. The absence of spurious vertical features in the comparisons we have presented here underlines the general much closer agreement amongst the more modern reanalysis systems."

p13, L18-19: I am not so familiar with the CALIOP classification of PSCs. Do MIX1, MIX2, and MIX2-enh include or are mainly dominated by NAT?

All the CALIOP MIX classifications include NAT (detected by the above threshold non-zero depolarization signal) and the ubiquitous STS. Although STS may be the dominant component, whenever NAT is present (above the detection threshold) the classification is reported as a MIX type. From simulations (see Fig 10, Pitts et al 2013) the NAT number densities are $<10^{-3}$ cm$^{-3}$ for MIX1, $10^{-3}$ to $10^{-1}$ cm$^{-3}$ for MIX2 and $>10^{-1}$ cm$^{-3}$ for MIX2-enh. The STS classification therefore means that no NAT is detected above the detection threshold.

p14, L19-p15, l2: Section 4.6 does not seem to present any results, but recapitulates the method? Maybe merge with Sect. 4.7?

Section 4.6 was a summary of the procedure we followed. Following the suggestion of referee RC2 we moved the text to the methods section.

p16, L13-16: I would not directly expect the distance of reanalysis grid points to observations being a reason for causing two distinct modes of temperature precision because I would expect those distances to vary continuously over the study area?

Satellite observations would drift through all grid points as you indicate. However, against this background there may be radiosonde observations recurring in practically static locations close to the grid points.

p17, L15-16: I am afraid, do not fully understand this sentence: "The standard deviations result from the combination of the quadrature addition of the precisions of the reanalyses and the reference points. "What does 'precision' of the reference points" refer to?

We mean the precision of the reference point temperatures as estimated in Figs 1c and 2c.

p18, L24-25: However, the standard deviations with respect to the reanalyses are only similar, because gravity wave-affected profiles showing small-scale temperature fluctuations have been excluded in your analysis, right? It might be good to repeat this here.

This is correct, but the data with detected gravity waves constitutes only about 4% and 12% for SH and NH respectively as stated in the text. Excluding these small fractions probably does not reduce the standard deviations much. We indicated in the revised text that we are effectively applying a low-pass filter to examine the large-scale temperature structure.

p19, L8-9: I would agree that this study showed much better agreement between the reanalyses and observations than earlier work. However, I am not sure if you really demonstrated that point in the paper (see also comment regarding p13, L11-15)? The discussion of this aspect could be extended a bit.

This is now discussed at the end of the results section.

p31, Fig. 3: Maybe indicate the noise variance of the COSMIC data in Figs 3a and 3d? In the caption the unit "$K^2$" is missing in a few places, e.g. "0.1" $->$ "0.1 $K^2$" and so on.

The estimated noise variance is about 0.03 $K^2$. We also fixed the units in the caption.

p33, Fig. 5: Does this analysis consider filtering of small-scale fluctuations due to gravity waves? There seem to be notable differences in the medians between the reanalyses. Is this related to COSMIC data being or not being assimilated in some of the reanalysis? How do these results compare to other studies?

Yes, the same gravity wave threshold as in Fig 3 is applied here. We indicate this at the end of the section on the wave case study. Only MERRA does not assimilate GPS-RO and this does have an appreciable effect on the median values in the SH and NH. We find little variation of the medians and standard deviations over the latitude range and comment on this at the end of the results section.

Technical Corrections

p11, l4: Remove units "J $Kg^{-1}$" from equation (3)? And "Kg" should be "kg" in other places?

Ok.

p12, L10: peninsula − > Peninsula (maybe use either "Palmer Peninsula" or "Antarctic Peninsula" throughout the manuscript for consistency?)

Palmer is now used throughout.

p12, L11: lambda_v − > lambda_z

Ok.

p14, L31: ERA-I − > ERA-Interim (and in other places)

We prefer to use ERA-I in the text for consistency with the figures and therefore instead substituted ERA-Interim − > ERA-I

p36, Fig. 8 and p37, Fig. 9: y-axis labels are partly missing

The 2nd and 3rd rows are successive enlargements of a region in the 1st row of images centered at 32 hPa. The x by y dimensions in km are given in the text. y-labels indicating the pressure would not add useful information.

p44, Fig. 16: fix "standard (triangles) deviations"

The caption now reads "Median deviations (diamonds) and standard deviations (triangles)..."

**RC2 General Comments:**

The authors present an evaluation of stratospheric temperatures at high latitudes in reanalyses by using temperatures from different satellite based observations including Microwave Limb Sounder (MLS) and GPS radio occultation (RO) measurements. They furthermore use observed regimes of different types of Polar Stratospheric Clouds (PCSs) for deriving temperature references based on the thermodynamics of supercooled ternary solutions and ice clouds.

The paper is well written but overall it is quite lengthy and includes 21 figures. At several places the content could easily be condensed and streamlined without losing information and thus presented in a more compact way. A sincere restructuring of the text in some sections is recommended and a restructuring/merging of figures. Furthermore, clarifications are needed regarding the definition of precision and accuracy, the choice of pressure levels, and the description of comparison methods (please see comments below).

I recommend publishing the manuscript after revision taking into account the reviewers recommendations and providing necessary clarifications. Please find a list of major and minor comments below

We will address the concerns of the referee in the modified manuscript. We reply to specific comments below.

**RC2 Major Comments:**

Abstract: The abstract is much too long and definitely should be shortened by half. Only essential information and results should be stated. E.g., line 10 to 16 on page 2 could be deleted. Is the statement on GPS RO uncertainties correct: "accuracy < 0.2 K, precision > 0.7K" (also in paper text and conclusion section)? To my knowledge it should be vice versa. The precision is about 0.25 K and the accuracy in the tropopause region about 0.7 K increasing exponentially in the stratosphere (see also comment below).

The abstract has been shortened to a reasonable length.

Your comments on accuracy vs precision for GPS-RO appear to be incorrect according to the literature on the RO measurement technique, which we cite in some detail on pages 7 and 8. We note that it is the precision (standard deviation) that increases exponentially in the stratosphere, whereas the accuracy (bias) is almost constant. Figs 18f and 19f indicate very good agreement with the standard GPS RO model error given in Eqn 1.

Methods:

Computation of pressure levels:
"The synoptic gridded reanalysis temperatures are interpolated to a common vertical pressure grid (100-10 hPa), with 6 levels per decade ..." Why are the pressure levels chosen this way for the reanalyses and the radio occultation observations, e.g., levels of 61 hPa or 46 hPa, etc.? Is there a specific reason for this? I strongly recommend using standard pressure levels or pressure levels with 10 hPa steps, i.e., plain numbers, as used later in the manuscript anyway. What

does "fixed pressure levels per decade" mean? "For convenience we interpolated the COSMIC temperature profiles to a set of fixed pressure levels with 24 levels per decade ..., approximately 2/3 km vertical resolution." Please revise the last statement.

The "odd looking" pressure levels are a subset of those chosen by the Aura science team for all instruments on the Aura platform. The pressure levels, $p(i)$, are defined as a fixed number of levels, $n$, per decade change in pressure (i.e. $p(i) = 10^{i/n}$) so they are in a logarithmic series. This is necessary because the MLS measurements cover a large region of the atmosphere and therefore linear 10 hPa pressure steps are not practical for the MLS instrument.

The MLS data processing retrieves products on either a low resolution $n = 6$ levels per decade pressure grid (e.g. for HNO3, N2O) or higher resolution $n = 12$ levels per decade pressure grid (e.g. for temperature, H2O, O3). Note that the levels have a structure similar to the well-known E-series (E6 and E12) of preferred values for electrical components such as resistors.

Since GPS-RO and reanalyses are of higher vertical resolution than the MLS grids, it is appropriate to interpolate both of these to the MLS low resolution pressure grid. We use the 6-level per decade grid since our definition of thermodynamic reference temperatures is limited by its dependence on the low resolution HNO3 data.

In the gravity wave case study. We use the native NCEP-CFSR reanalysis levels 100, 70, 50, 30, and 20 hPa since the complications of the CALIOP/MLS field of view are not involved in this section. The MLS H2O data are instead interpolated to the reanalysis grid.

The vertical resolution of GPS RO observations ranges from a 0.5 km (100 m in case of wave optics processing) in the troposphere to about 1.5 km in the stratosphere. By interpolating the vertical levels you change the sampling of the height/pressure levels.

The atmPrf GPS-RO temperatures are supplied on a vertical grid with spacing of about 20m. In the separate analysis used for gravity waves the GPS-RO data are interpolated to a high resolution pressure grid (24 levels per decade) with approximately 2/3 km vertical spacing.

Section 2.4:

Please explain explicitly how the reanalyses are compared with the radio occultation observations. As I understand you compare the gridded reanalyses data with individual COSMIC temperature profiles (supposedly averaged in the respective grid or zonal mean). Doing it this way you would not account for the different sampling of the observations. A comparison should be either performed by subsampling the reanalyses at the locations of the RO observations (i.e., comparing colocated profiles) or by comparing monthly-mean zonal-mean reanalyses data with zonal-mean sampling error corrected RO climatologies.

We do actually compare colocated profiles. Unfortunately, we omitted this important detail of the reanalysis vs COSMIC comparisons from the original manuscript. The sampling of the reanalysis gridded data for the GPS-RO measurements is done the same way as for the MLS temperatures as indicated in Section 2.1 ...

"The synoptic gridded reanalysis temperatures are interpolated to a common vertical pressure grid (100–10 hPa), with 6 levels per decade ($p = 100 \times 10^{i/6}$; $i = 0, 1, \ldots, 6$), and to the MLS measurement times and geophysical locations."

This creates a colocated set of reanalysis and MLS profiles. A different set of colocated profiles is created from the reanalysis and COSMIC data. We amended the text to reflect the similar treatment of GPS and MLS. Note that we do not try to obtain colocated MLS and GPS profiles.

Cite Schreiner et al. (2007; doi:10.1029/2006GL027557) in section 2.4 as they made the first analysis on the precision of COSMIC data. Their results were confirmed by Alexander et al. (2014), who derived a precision of about 0.1% in refractivity between 8-25 km (larger above 30 km) which translates to about 0.25 K. No strong dependence of the estimated precision on latitude or season was found.

We added the citation to Schreiner et al. "Schreiner et al (2007) investigated the precision of the COSMIC data by leveraging the close configuration of the six satellites during the early deployment phase of the mission." We have already cited Alexander et al and others in the same section. Also from Alexander et al., we noted that between 10 and 30 km the temperature bias was found to be +0.1 K i.e. almost a constant value for accuracy with height.

Is this consistent with your statement in the abstract and conclusion section on precision and accuracy?

Yes.

Please include a definition of precision and accuracy in the method section before values for the different data sets are discussed and give an explicit description on how precision and accuracy are computed in your study.

We use standard definitions of accuracy and precision. (i) high accuracy is determined by the narrowness of the spread of repeated measurements relative to the underlying true value. (ii) high precision is determined by the narrowness of repeated measurements with respect to a measure of their central value such as the mean or median. Therefore the mean of a set of precise measurements may have a large deviation (bias) from the true value, although the standard deviation (about the mean) of the precise set may be small. High accuracy measurements have by definition a small bias and small standard deviation.

In this study we use the GPS-RO temperatures because of their previously validated high accuracy and precision. We also use the reference temperatures calculated from the thermodynamics of PSC formation.

Details are given in the "Datasets and methodology" section...

"The accuracy (bias) of the reanalysis temperatures are obtained relative to the reference temperatures (`LIQ`, `ICE`, and COSMIC). We use the calculated standard deviations of the temperature differences to estimate the measurement precisions i.e. high precision is determined by the narrowness of repeated measurements with respect to a measure of their central value such as

the mean or median."

Move method descriptions from the results section to the method section (see comments below on restructuring).

All suggestions on restructuring the manuscript have been followed where possible.

**Structure of the manuscript:**

Introduction:

Page 4, 3rd paragraph: This paragraph already describes the methodological approach in some detail and would rather fit at the beginning of the method section.

Ok.

Page 4, line 29: "However, lack of sufficient statistics in the Arctic precludes a robust conclusion for ICE PSCs." This is a result and the sentence should be moved to the conclusion section or/and the results section.

Ok.

Page 4, line 31-33: Detailed description of investigated dates should be moved to the beginning of the methods section.

Ok.

**Method section:**

I recommend slight restructuring: Start with the reanalyses description. The information in the first paragraph does not really fit here at the beginning and should be moved to sections 2.2 where CALIOP observations are described and to section 2.3 where MLS is described.

Ok.

Section 2.1: The information on all the reanalyses could be summarized in a Table and would give a more compact overview.

A table has been provided.

Streamline page 7 and 8 as you recapitulate several studies in quite a lengthy way.

We prefer to retain this level of detail.

Results sections:

The results sections include method descriptions such as section 4.6 describing how the comparison of reanalyses temperatures to LIQ and ICE reference points is performed and in section 4.8.2 (last paragraph) on difference computations. These method descriptions should be moved to the method section.

Ok.

Furthermore some results sections are very short and could be merged into one section such as sections 4.4, 4.5, and 4.7. There is no need to make a separate section for each figure but rather to comprise related topics in one section. The merged section 4.4 could be entitled, e.g., "PSC types from CALIOP and their representation in MLS observations".

Ok.

Section 4.8 Polar temperature reference points:

Remove sub-sub-sections and streamline the text.

Ok, minor sub sections were removed.

Section title could be "Reanalysis temperatures compared to LIQ and ICE temperature reference points"

Ok.

Figures:

General: You often provide the same information twice in the figure captions and in the description of figures in the text. Please give the technical description of the figure in the caption and provide explanations and description of the scientific results in the text, e.g., Fig. 13.

Some information is repeated in the caption so the reader does not have to flip back and forth between the text and the figure.

Fig. 5: The latitudinal structure from 60S/N to 90S/N is rather uniform for the median and the standard deviation. A structure in temperature differences is mainly visible in the vertical domain. Averaging over 60S-90S and 60N-90N would give a condensed overview. Median and standard deviation profile could be shown in one panel. Fig. 5 could be condensed in 2 rows for the Arctic and Antarctic. In Figure 6 the information on latitudinal differences and vertical differences is given again, anyway.

We prefer to retain the format of Fig 5. As you note, the structure is immediately visible in this presentation format. In addition we now refer to this figure in the discussion of comparisons with a recent paper by Hoffmann et al. (see response to RC1 above).

Fig. 6: The vertical boxes should also be centered around a zero line as done for the latitudes.

There is no zero-line for the vertical scale.

Fig. 9: Please condense information. Fig.9a is the same Figure as already shown in Fig. 8. Fig.9b does not really give much information, it can be stated in the text. Please check whether this figure is really needed or the information could be stated in the text.

Fig. 9 has been deleted. Essential information was placed in the text instead.

Fig. 11: What is the reason for showing the information twice? Why do you show inverse temperature? It is distracting to the reader unless there is a specific reason for it. I suggest plotting a logarithmic y-axis, using temperature for the x-axis, and showing only 2 plots in Fig. 11.

We believe it is helpful to show both diagrams (a) because of the continuity with Fig 2 and (b) because of the high degree of linearization that results from the transformation to log mixing ratio vs 1/T (see Murphy and Koop, QJRMS (2005), doi: 10.1256/qj.04.94).

Fig. 12: Suggest having a similar design as for Fig. 11, i.e., plotting the temperature axis from 175 K to 200 K and removing the inverse temperature x-axis.

Again, this plot format is used because log(PICE) vs 1/T has a linear relation (black/blue dashed line).

Fig. 18 and 19: The standard deviations (Fig. 18.d-f) could be easily plotted in Fig. 18 a-c by extending the x-axis to -3K to 3K. The same could be done for Fig. 19. Fig. 18 and 19 could then be merged to one Fig. showing SH in the upper panels and NH in the lower panels. So the reader would be provided with compact information.

The figures were designed to be quite easy to read in their current format and we prefer not to squash the space along the x-axis in order to cram more plot lines together.

Fig. 14 and 15: Please compile the information printed in the different panels in a Table. Fig. 14 and 15 could be removed and information of section 4.8.1 could be streamlined. The description at page 16, line 10 to 18, could be removed and the main information stated in one sentence.

Figures 14 and 15 were removed. The information is now supplied in Tables 1 and 2. The description on the possibility of more than one mode has been retained.

Minor comments:

The title is very long and contains 5 acronyms. It should be shortened. I think it is not necessary to list all the sensors explicitly in the title, could be replaced by "satellite observations".

We prefer to retain the acronyms in the title as they provide simple built-in keywords for search engines whereas "satellite observations" is far too vague.

Also the term "supercooled ternary solutions" might not be a familiar term for many readers. Maybe use "polar stratospheric cloud types or polar stratospheric cloud regimes" instead.

We prefer to retain the kind of PSCs in the title. "Supercooled ternary solution" has been in use for over 30 years.

Reanalysis: More than one reanalysis is evaluated, rather use "reanalyses" in the title.

We have modified the title. "Accuracy and precision of polar lower stratospheric temperatures from reanalyses evaluated from A-train CALIOP and MLS, COSMIC GPS RO, and the equilibrium thermodynamics of supercooled ternary solutions and ice clouds."

Page 2, line 19: "significant low bias in MLS temperature of up to 3 K" ? Do you mean "large bias"?

No, we mean low bias. MLS temperatures are lower by up to 3 K.

Page 2, line 31: What does "(one-off research satellites)" mean, check singular/plural.

We changed "one-off" to "one-of-a-kind". This refers to any kind of satellite instrument (such as MLS, MIPAS etc) that is not part of a long-term dedicated series of operational satellites (such as NOAA GOES).

Page 2, line 32: What is "temperatures measured with the Geophysica": Please be explicit. What is the Geophysica, how were the temperatures measured, state the instrument?

The Geophysica is a high-altitude research aircraft and ambient temperatures were measured by the Thermodynamic Complex instrument (Shur et al., 2006). We do not see the need to include this reference.

"Wegner et al (2012) showed a comparison of ERA-I Arctic temperatures in March 2005 that indicates a 1.5 K warm bias in ERA-I below 205 K compared to the ambient temperatures measured with the Geophysica high-altitude research aircraft."

Page 6, line 26: "-2-0 K" change to "-2 K to 0 K"

Ok.

Page 7, line13: "requires a number of assumptions because of the long ray path through a nonuniform atmosphere. Therefore, corrections are required for ionospheric effects, variations in water vapor, and gradients in in temperature along the ray path." This statement is not fully correct. Please reformulate/clarify.

Initialization (of bending angles) at high altitudes is performed where the signal to noise ratio of the measurement becomes low. An ionospheric correction is applied through differencing of the two GPS frequencies to remove the ionospheric contribution to the measurement and to gain information on the neutral atmosphere only. Refractivity then includes information on the dry and moist part of the atmosphere (see Smith and Weintraub formula, as you explain on the next page). In a dry atmosphere, temperature can be retrieved without any further correction or initialization. In a moist atmosphere, i.e., in the mid- to lower troposphere, additional background information is needed to separate temperature and humidity information (usually through a 1DVar retrieval).

The above description does not appear to identify what is not "fully correct" in our original statement and therefore we have not modified it.

However, as you investigate temperatures at high latitudes in the stratosphere, you can use dry temperatures only. The discussion on wetPrf temperatures can be removed as they are not used.

Others papers have used wetPrf for their studies and in fact we have repeated the entire analysis using the wetPrf data. We have also compared the COSMIC version 2013.3520 wetPrf and atm-Prf temperatures and have therefore reported their differences. The point is that the differences are small enough that either could have been used for this analysis as is now noted in the revised manuscript.

Page 7: line 34: Remove "similar results were obtained over data sparse (Pacific) and data dense (USA) regions". This is distracting as one might think it is RO data sparse and dense regions (but RO has almost equal distribution globally) but actually it relates to observations assimilated into ECMWF.

Ok, removed.

Page 8, line 21,22: insert "about" before "14 km" and before "9 km".

Ok, inserted.

Page 9, line 20: Why a pressure range of 68 to 21 hPa and not 70 to 20 hPa.

As discussed previously the pressure range quoted is for our standard MLS pressure levels.

Page 10, Eq.3: correct to "J kg-1" and through paper text.

Ok, corrected.

Page 11: "A decrease in the spread of the temperature differences of reanalysis data compared to COSMIC should be detectable by rejecting the temperature profiles from consideration that are expected to have lower fidelity to the true atmosphere." What is meant by fidelity to the true atmosphere? The truth is actually not known. Rejecting observational profiles with higher variance will of course improve the difference between reanalyses and observations if observations show a higher variance than the reanalyses at high latitudes, which is well known for RO. I actually do not understand the purpose of rejecting profiles with higher variance.

"The truth is actually not known." Quite true. However, we are able to reject specific reanalysis temperature profiles that are likely to be poorly represented because the model resolution cannot accurately capture gravity wave effects. We postulate that the remaining sample of reanalysis temperatures will be closer to the true temperatures (i.e. higher fidelity). We generate a proxy

for gravity wave activity at each GPS-RO location from the variance in the vertical structure of the COSMIC temperatures (80–20 hPa). If the variance is above a chosen threshold then we reject the matched reanalysis/COSMIC profile from consideration. We can see from Fig 3(c,f) that both the variance and mean difference between ERA-I and COSMIC are improved using this method. Fig 3(b,c) shows that this is not just some isotropic removal of "bad" high variance profiles at high latitudes since the geophysical structure reveals an association with known gravity wave "hot-spots".

We have modified the explanation as also suggested by referee RC1.

Page 12, line 28: What do you mean with "the geophysical structure"? The vertical structure versus the latitudinal structure at high latitudes is shown.

Removed the word "geophysical".

Page 13, line 2: Use clear wording in description of Fig. 6. Rather than saying the "box heights" it is the differences in the vertical between reanalyses and RO.

The terms box width and box height are defined on lines 6–7 of Page 13, these are used to simplify the description.

Page 13, line 22: Why do you show the 46 hPa level?

It is a representative level. Other figures also show the 46 hPa level: Figs 11, 14, 15, 16, and 17. Fig 13 displays 31 hPa because the HNO3 range is larger than at 46 hPa.

Page 14, end of section 4.5: Please give percentage of scenes analyzed with respect to available scenes.

Only about 15% of the scenes containing STS or ice PSCs are accepted. This information was added to the text.

Page 17: The last part of section 4.8.2 (page 18, first paragraph) is very hard to read. Please reformulate and streamline info.

Yes, we agree the paragraph looks ugly, but we believe the math is correct.

Page 44, Fig. 16, caption: Correct to: "Median (diamonds) and mean (triangles)..." Section titles: Please remove Tre-Tro, Tre-Teq, and similar in section titles.

The caption now reads "Median deviations (diamonds) and standard deviations (triangles)..." Section titles were removed as part of the restructuring suggested above.

---

## Author Response (AR2)

**acpd-2017-640 Reply to Editor Comments**

**Accuracy and precision of polar lower stratospheric temperatures from reanalyses evaluated from A-train CALIOP and MLS, COSMIC GPS RO, and the equilibrium thermodynamics of supercooled ternary solutions and ice clouds**

**A. Lambert and M. L. Santee**

Editor Comments are in blue type.

Author Responses are in brown type.

**Co-Editor Decision: Publish subject to minor revisions (review by editor) (13 Dec 2017) by Gabriele Stiller**

**Comments to the Author:**
I have checked your responses to the reviewers' comment. In general, I am satisfied with your responses and related changes. However, there is one remaining issue that I would like you to give some attention:

Referee no. 2 has pointed to the definition of precision and accuracy in your paper. I your reply, you are not fully consistent. On page 12 of your reply, you provide a definition of precision and accuracy: "We use standard definitions of accuracy and precision. (i) high accuracy is determined by the narrowness of the spread of repeated measurements relative to the underlying true value. (ii) high precision is determined by the narrowness of repeated measurements with respect to a measure of their central value such as the mean or median." I understand that what you call accuracy is the combination of precision and bias (small bias AND small spread provides good accuracy). Later on, however, in your reply and in the manuscript, you use "bias" alone and "accuracy" synonymously (Page 12, bottom: "The accuracy (bias) of the reanalysis temperatures ...". Therefore I suggest that you start your paper with a clear and careful definition of the terms accuracy and precision as used in the paper, and stick with this definition throughout the paper.

**Reply to Editor Comments:**

We thank the editor for pointing out this discrepancy which we have resolved through a small number of changes in the manuscript detailed below. Please see the track changes document for further context.

Abstract. Lines 14 and 17:
accuracy → bias

Page 3, Line 3:
No changes to the text, apparently this is a result of changing the latex command to italicise from \it to \textit.

Page 5, Lines 15–21:

We added two bullet points in the methodology section to clarify our usage of *High precision* and *High accuracy*.

Page 7, Line 13:
Now both the bias and precision values are given.

Page 8, Lines 16–22:
The description of the GPS RO model error *precision* (Eq 1) has been separated from the preceeding discussion of the GPS RO biases by starting a new paragraph, we also clarified by adding the word *precision*.

Page 10, Lines 13, 14, 17 and 24:
accuracy → bias

**Accuracy and precision of polar lower stratospheric temperatures from reanalyses evaluated from A-train CALIOP and MLS, COSMIC GPS RO, and the equilibrium thermodynamics of supercooled ternary solutions and ice clouds**

Alyn Lambert[1] and Michelle L. Santee[1]

[1]Jet Propulsion Laboratory, California Institute of Technology, Pasadena, California, USA

*Correspondence to:* A. Lambert, (Alyn.Lambert@jpl.nasa.gov)

**Abstract.**

We investigate the accuracy and precision of polar lower stratospheric temperatures (100–10 hPa during 2008–2013) reported in several contemporary reanalysis data sets comprising two versions of the Modern-Era Retrospective analysis for Research and Applications (MERRA and MERRA-2), the Japanese 55-year Reanalysis (JRA-55), the European Centre for
5    Medium-range Weather Forecasts (ECMWF) interim reanalysis (ERA-I), and the National Oceanic and Atmospheric Administration (NOAA) National Centers for Environmental Prediction (NCEP) Climate Forecast System Reanalysis (NCEP-CFSR). We also include the Goddard Earth Observing System Model version 5.9.1 near real-time analysis (GEOS-5.9.1). Comparisons of these datasets are made with respect to retrieved temperatures from the Aura Microwave Limb Sounder (MLS), Constellation Observing System for Meteorology, Ionosphere and Climate (COSMIC) Global Positioning System (GPS)  Radio Occultation
10   (RO) temperatures, and independent absolute temperature references defined by the equilibrium thermodynamics of supercooled ternary solutions (STS) and ice clouds. Cloud-Aerosol Lidar with Orthogonal Polarization (CALIOP) observations of polar stratospheric clouds are used to determine the cloud particle types within the Aura MLS geometric field of view. The thermodynamic calculations for STS and the ice frost point use the colocated MLS gas-phase measurements of $HNO_3$ and $H_2O$. The estimated  bias and precision for the STS temperature reference, over the 68 to 21 hPa pressure range, is
15   0.6–1.5 K and 0.3–0.6 K, respectively; for the ice temperature reference they are 0.4 K and 0.3 K, respectively. These uncertainties are smaller than those estimated for the retrieved MLS temperatures and also comparable to GPS RO uncertainties (bias<0.2 K, precision >0.7 K) in the same pressure range.

We examine a case study of the time-varying temperature structure associated with layered ice clouds formed by orographic gravity waves forced by flow over the Palmer Peninsula, and compare how the wave amplitudes are reproduced by each
20   reanalysis data set. We find that the spatial and temporal distribution of temperatures below the ice frost point, and hence the potential to form ice PSCs in model studies driven by the reanalyses, varies significantly because of the underlying differences in the representation of mountain wave activity.

High-accuracy COSMIC temperatures are used as a common reference to intercompare the reanalysis temperatures. Over the 68–21 hPa pressure range, the biases of the reanalyses with respect to COSMIC temperatures for both polar regions fall

within the narrow range of −0.6 K to +0.5 K. GEOS-5.9.1, MERRA, MERRA-2 and JRA-55 have predominantly cold biases, whereas ERA-I has a predominantly warm bias. NCEP-CFSR has a warm bias in the Arctic, but becomes substantially colder in the Antarctic.

Reanalysis temperatures are also compared with the PSC reference temperatures. Over the 68–21 hPa pressure range, the reanalysis temperature biases are in the range −1.6 K to −0.3 K with standard deviations ∼0.6 K for the CALIOP STS reference, and in the range −0.9 K to +0.1 K with standard deviations ∼0.7 K for the CALIOP ice reference. Comparisons of MLS temperatures with the PSC reference temperatures reveal vertical oscillations in the MLS temperatures, and a significant low bias in MLS temperatures of up to 3 K.

The author's copyright for this publication is transferred to the California Institute of Technology.
Copyright 2017 California Institute of Technology. Government sponsorship acknowledged.

[revised manuscript text omitted]

---

## Author Response (AR3)

**acpd-2017-640 Manuscript was accepted for final publication in ACP:**

[revised manuscript text omitted]